# ALS-ActLR: Alternating Least Squares based Activation-Aware Low-Rank Model Compression

## Abstract

Large language models (LLMs) achieve state-of-the-art performance but remain impractical for on-device deployment due to memory and compute constraints, making compression essential. Activation-aware low-rank approximation is promising, yet existing methods follow a two-step *approximate-then-factorize* routine, which couples the factors and weakens preservation of salient activation structure. We present *ALS-ActLR*, which combines a spectral-informed metric transformation (SIMT) with *Activation-aware alternating least squares (ALS)* to optimize the low-rank factors directly using a tiny calibration set (256 samples). A subsequent uncertainty-weighted distillation stage further recovers lost information by adaptively balancing cross-entropy, knowledge distillation, and feature alignment. Experiments show that *ALS-ActLR* substantially reduces parameters and floating-point operations (FLOPs) while preserving accuracy and perplexity, consistently outperforming strong baselines. Concretely, on Llama-7B at 60% compression (i.e., 60% parameters removed, 40% retained), it reduces mean perplexity to 27.74 (a 69.0% reduction over the best baseline) and raises accuracy to 48.92% (+3.72 points), while achieving the best scores across 40–80% compression and a wide range of model scales from 1.1B to 13B across multiple families. These results highlight *ALS-ActLR* as a scalable and effective framework for activation-aware compression.

## 1 Introduction

Large language models (LLMs) attain state-of-the-art results across diverse tasks (Gozalo-Brizuela & Garrido-Merchán, 2023; Zhao et al., 2025), yet their parameter counts, activation footprints, and FLOP requirements remain incompatible with the memory, bandwidth, and power budgets of edge platforms (Wan et al., 2024; Wang et al., 2024; Zhou et al., 2024). As a result, principled compression is essential for on-device deployment (Zhu et al., 2024). Existing paradigms include quantization (Frantar et al., 2023; Lin et al., 2024; Xiao et al., 2024), pruning (Frantar & Alistarh, 2023; Ma et al., 2023), knowledge distillation (KD) (Gu et al., 2024; Hsieh et al., 2023; Ben Noach & Goldberg, 2020; Sy et al., 2025), and low-rank approximation (LRA) via matrix factorization (Hsu et al., 2022; Yuan et al., 2024; Wang et al., 2025b). Unlike hardware-dependent quantization and pruning that often require expensive large-scale retraining, weight-only LRA can be data-free, and post-training KD typically operates with a small calibration set, yielding a lightweight, hardware-agnostic pipeline (Ben Noach & Goldberg, 2020; Sy et al., 2025). Recent activation-aware approaches—which tailor the approximation to the distribution of layer activations—consistently outperform naive weight-only LRA by better preserving output fidelity (Wang et al., 2025b; Sy et al., 2025). However, prevailing SVD-based activation-aware methods adopt a two-step procedure that first constructs a surrogate low-rank weight $W'$ by minimizing $\min_{W'}\|WX - W'X\|$ and then factorizes $W'$; this coupling of factors hinders preservation of salient activation patterns and limits its effectiveness under tight resource constraints (Hsu et al., 2022; Yuan et al., 2024; Wang et al., 2025b; Sy et al., 2025). In addition, conventional multi-objective KD commonly relies on fixed loss weights, lacking noise-aware adaptive balancing across heterogeneous objectives and thus impeding well-calibrated optimization (Ben Noach & Goldberg, 2020).

However, these limitations call for a direct activation-aware factorization that avoids surrogate coupling and a principled update stage that balances heterogeneous objectives under calibration noise. Our approach, *ALS-ActLR*, addresses these needs with the following contributions:

- **Algorithmic framework.** We propose *ALS-ActLR*, an activation-aware *LRA–then–update* framework that unifies spectral-informed metric transformation (SIMT), activation-aware alternating least squares (ALS), and uncertainty-weighted multi-objective distillation (UW-MOD). Unlike prior two-step SVD methods, *ALS-ActLR* directly minimizes activation-weighted error with decoupled factors and adaptively balances heterogeneous losses.

- **Theoretical foundations.** We establish that the activation-aware ALS subproblems are strongly convex with unique solutions, prove sufficient descent of the global objective, and show convergence of the iterates to a stationary point under standard assumptions. These results ensure the method is not only empirically effective but also theoretically well-grounded.

- **Empirical validation.** Extensive experiments across multiple compression ratios, model families, and backbone sizes show that *ALS-ActLR* consistently achieves lower perplexity and higher accuracy than existing *LRA–then–update* approaches. Figure 1 highlights simultaneous improvements in both metrics, confirming robustness and scalability.

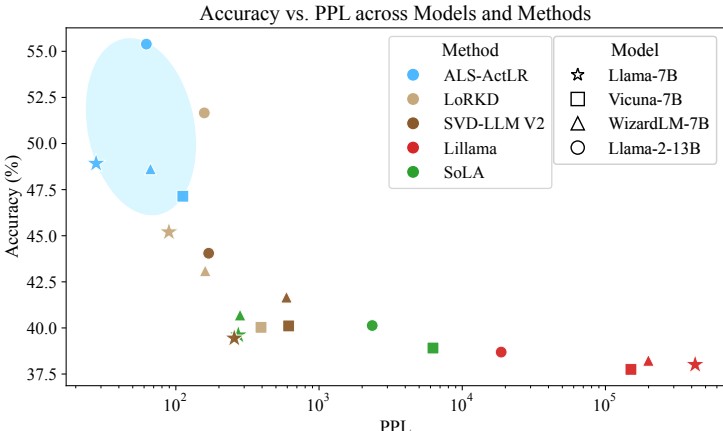

Figure 1: Perplexity–accuracy comparison across models and methods at 60% compression with 256 calibration samples. ALS-ActLR consistently achieves lower perplexity and higher accuracy, demonstrating robust improvements across backbones.

## 2 RELATED WORK

The classical singular value decomposition (SVD) factorizes a matrix into orthogonal bases and singular values, with truncation yielding the best low-rank approximation under Frobenius error (Golub & Van Loan, 2013). For LLM compression, truncating the SVD of layer weights is a natural baseline, but it minimizes parameter error rather than preserving activation fidelity (Ben Noach & Goldberg, 2020). To incorporate activation information, ASVD rescales columns using activation statistics before SVD (Yuan et al., 2024); SVD-LLM applies truncation-aware whitening via a Cholesky factor of the activation covariance, providing a direct mapping from discarded singular values to loss (Wang et al., 2025b); and SVD-LLM V2 further reduces truncation loss by allocating heterogeneous compression ratios across weight matrices according to their theoretical contribution to $\|WX - W'X\|_F^2$ and replacing Cholesky whitening with a more numerically stable two-stage SVD-based truncation that still attains the optimal Frobenius loss (Wang et al., 2025a). Lillama decorrelates activations with a covariance root and then applies SVD to the reweighted weights (Sy et al., 2025). Beyond such weight-centric designs, SoLA exploits "soft activation sparsity" in FFNs by keeping a small set of highly activated neurons dense while applying SVD-based low-rank decomposition with component-wise rank allocation to the remaining channels, yielding strong training-free compression across LLaMA and Mistral models (Huang et al., 2025). Despite these advances, all SVD-based activation-aware methods follow a two-step process: constructing a surrogate

low-rank weight matrix $W'$ (possibly after activation-aware reweighting or neuron partitioning) and then factorizing it into $A$ and $B$. This coupling obscures salient activation patterns, reduces compression effectiveness, and limits applicability under resource constraints. To our knowledge, no prior method directly minimizes $\|(W - AB)X\|_F^2$ by optimizing $A$ and $B$ jointly.

The *LRA–then–update* framework, viewed as a post-approximation recovery mechanism, leverages a small calibration set to re-optimize the compressed model and recover accuracy lost after the initial low-rank factorization (Ben Noach & Goldberg, 2020; Wang et al., 2025b; Sy et al., 2025). In the teacher-driven variant, the compressed model is adapted to match the teacher's intermediate and output distributions while also fitting calibration labels (Sanh et al., 2020; Lewis et al., 2020). In the student-driven variant, the teacher is conditioned on the student's activations and the student aligns its layerwise outputs with the teacher (Wang et al., 2025b; Sy et al., 2025; Yu & Wu, 2023). Due to direct supervision from high-fidelity teacher activations, the teacher-driven approach generally converges faster. However, both variants rely on fixed loss weights, and mismatched objective scales hinder balanced optimization, often yielding suboptimal convergence.

In summary, existing activation-aware SVD pipelines—including ASVD, SVD-LLM and SVD-LLM V2, Lillama, and SoLA—reshape weights using activation statistics or neuron-wise importance signals and then factorize a surrogate matrix, a two-stage coupling that suppresses salient modes. *LRA–then–update* methods can partly recover accuracy but still rely on fixed loss weights, leading to poorly balanced objectives. These limitations motivate a direct activation-aware factorization that jointly optimizes $A$ and $B$ with respect to activations, while enabling data-driven rebalancing in the update phase. We next introduce *ALS-ActLR*, which embodies this idea.

# 3   ALS-ACTLR

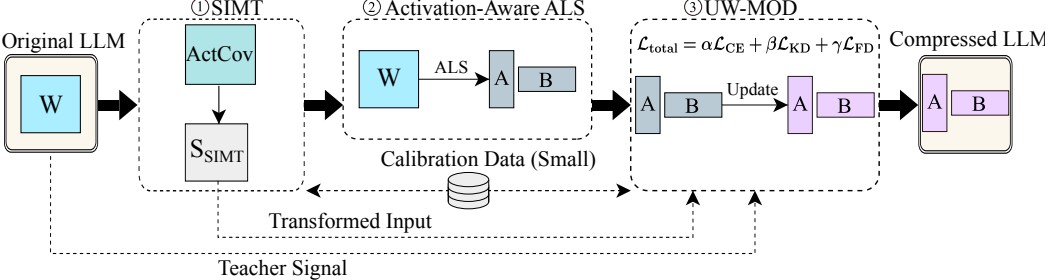

Figure 2: Overview of ALS-ActLR: SIMT reparameterizes the objective by leveraging the spectral factor $S_{\text{SIMT}}$ of the activation covariance (ActCov) computed from 256 calibration samples; based on $S_{\text{SIMT}}$, activation-aware ALS factorizes each original linear-layer weight $W$ into low-rank matrices $A$ and $B$; and UW-MOD balances heterogeneous losses to recover compression accuracy.

Figure 2 overviews *ALS-ActLR*, an activation-aware framework for compressing LLMs within an *LRA–then–update* paradigm. The method comprises three mutually dependent components that jointly address the limitations of prior approaches. First, we construct a small calibration set by randomly sampling sentences (256 samples), collect the corresponding activation covariances, and apply the spectral-informed metric transformation (SIMT), which uses the spectral factor to reparameterize the objective into an equivalent weighted form. Second, we apply an activation-aware ALS procedure to the weight matrices to obtain a compact low-rank factorization. Finally, we refine the compressed model via uncertainty-weighted multi-objective distillation (UW-MOD), which adaptively balances heterogeneous supervisory signals. Formal definitions appear below, and the overall procedure is summarized as Algorithm 1 in Appendix A.2.

## 3.1   SPECTRAL-INFORMED METRIC TRANSFORMATION (SIMT)

Given a linear weight $W \in \mathbb{R}^{m \times n}$ and activation samples $X \in \mathbb{R}^{n \times N}$, the activation-aware low-rank approximation seeks $A \in \mathbb{R}^{m \times r}$ and $B \in \mathbb{R}^{r \times n}$ ($r \ll \min\{m, n\}$) that minimize

$$\min_{A,B} \|(W - AB)X\|_F^2 \;=\; \|W - AB\|_M^2, \qquad M := XX^\top. \tag{1}$$

Let $M$ be estimated on a small calibration set and admit the eigendecomposition $M = U_s \Lambda_s U_s^\top$. Define the *spectral factor* $S_{\text{SIMT}} := U_s \Lambda_s^{1/2}$, which satisfies $S_{\text{SIMT}} S_{\text{SIMT}}^\top = M$. Using $\|Z\|_M^2 = \|ZS\|_F^2$ for any $S$ with $SS^\top = M$, Eq. (1) is exactly equivalent to

$$\min_{A,B} \| WS_{\text{SIMT}} - ABS_{\text{SIMT}} \|_F^2. \tag{2}$$

Thus, SIMT reparameterizes the metric via a covariance square root, yielding a reweighted objective Eq. (2) that is equivalent to the original activation-weighted formulation. In practice, $M$ can be estimated by *streaming* second-moment accumulation over tokens in the calibration corpus (e.g., $M \leftarrow \frac{1}{N_c} \sum X_b X_b^\top$ for mini-batches $X_b$), which *avoids explicitly storing or manipulating the full activation matrix $X$*. Since $N$ equals the number of calibration examples times their token counts ($N \approx N_c T$) and greatly exceeds the hidden dimension $n$, this construction reduces memory traffic and *improves numerical conditioning and stability* in the subsequent least-squares subproblems.

## 3.2 ACTIVATION-AWARE ALS

We augment Eq. (2) with a ridge penalty $\frac{\tau}{2}\|A\|_F^2 + \frac{\tau}{2}\|B\|_F^2$ for conditioning and generalization, add proximal terms for stability, and optimize the objective via ALS. At each iteration, the two block subproblems decouple into convex quadratics:

$$A^{(t+1)} \leftarrow \arg\min_A \tfrac{1}{2}\|WS - AB^{(t)}S\|_F^2 + \tfrac{\tau}{2}\|A\|_F^2 + \tfrac{\rho}{2}\|A - A^{(t)}\|_F^2, \tag{3}$$

$$B^{(t+1)} \leftarrow \arg\min_B \tfrac{1}{2}\|WS - A^{(t+1)}BS\|_F^2 + \tfrac{\tau}{2}\|B\|_F^2 + \tfrac{\rho}{2}\|B - B^{(t)}\|_F^2. \tag{4}$$

Both admit efficient closed-form solutions: the $A$-step reduces to solving a small $r \times r$ linear system via Cholesky factorization, while the $B$-step simplifies into elementwise updates after a basis rotation. This keeps each iteration lightweight and stable. Full derivations are deferred to Appendix A.3, and the activation-aware ALS procedure is summarized as Algorithm 2 therein.

**Theoretical properties.** We analyze the base objective together with its proximal block models in the context of ALS. Throughout this section, the spectral factor $S := S_{\text{SIMT}}$ is fixed, and $\tau \geq 0$ (ridge) and $\rho \geq 0$ (proximal) denote fixed regularization parameters. At each iteration, we perform exact block minimization of the following models:

$$\Phi(A, B) := \tfrac{1}{2}\big\| WS - ABS \big\|_F^2 + \tfrac{\tau}{2}\big(\|A\|_F^2 + \|B\|_F^2\big), \tag{5a}$$

$$Q_A\Big(A \,\Big|\, A^{(t)}, B^{(t)}\Big) := \tfrac{1}{2}\big\| WS - AB^{(t)}S \big\|_F^2 + \tfrac{\tau}{2}\|A\|_F^2 + \tfrac{\rho}{2}\big\|A - A^{(t)}\big\|_F^2, \tag{5b}$$

$$Q_B\Big(B \,\Big|\, A^{(t+1)}, B^{(t)}\Big) := \tfrac{1}{2}\big\| WS - A^{(t+1)}BS \big\|_F^2 + \tfrac{\tau}{2}\|B\|_F^2 + \tfrac{\rho}{2}\big\|B - B^{(t)}\big\|_F^2. \tag{5c}$$

For clarity, we summarize the main results below and defer all proofs to Appendix A.4.

**Theorem 1** (Convexity and uniqueness of block subproblems)**.** *Fix $B$. The $A$-subproblem $\min_A Q_A(A \,|\, A^{(t)}, B)$ is a convex quadratic with Hessian*

$$\nabla_A^2 Q_A = (HH^\top) \otimes I_m + (\tau + \rho)I_{mr}, \qquad H := BS.$$

*Fix $A$. The $B$-subproblem $\min_B Q_B(B \,|\, A, B^{(t)})$ is a convex quadratic with Hessian*

$$\nabla_B^2 Q_B = (SS^\top) \otimes (A^\top A) + (\tau + \rho)I_{rn}.$$

*Both subproblems therefore admit closed-form minimizers. Moreover, they are strongly convex—and hence uniquely solvable—whenever $\tau + \rho > 0$. If $\tau = \rho = 0$, uniqueness holds under the rank conditions $\text{rank}(BS) = r$ for the $A$-step and $A^\top A \succ 0$, $SS^\top \succ 0$ for the $B$-step.*

**Theorem 2** (Sufficient descent of the base objective)**.** *Let $\{(A^{(t)}, B^{(t)})\}_{t \geq 0}$ be generated by Algorithm 2 with $\rho \geq 0$. Then for all $t$,*

$$\Phi(A^{(t+1)}, B^{(t)}) \leq \Phi(A^{(t)}, B^{(t)}) - \tfrac{\rho}{2}\|A^{(t+1)} - A^{(t)}\|_F^2,$$

$$\Phi(A^{(t+1)}, B^{(t+1)}) \leq \Phi(A^{(t+1)}, B^{(t)}) - \tfrac{\rho}{2}\|B^{(t+1)} - B^{(t)}\|_F^2.$$

*Consequently, $\{\Phi(A^{(t)}, B^{(t)})\}$ is non-increasing and convergent (bounded below by 0). If $\rho > 0$, then $\sum_{t=0}^{\infty}\|A^{(t+1)} - A^{(t)}\|_F^2 < \infty$ and $\sum_{t=0}^{\infty}\|B^{(t+1)} - B^{(t)}\|_F^2 < \infty$.*

**Theorem 3** (Stationarity of limit points). *Assume either (i) $\tau > 0$ (ridge regularization, which makes $\Phi$ coercive) or (ii) the sequence $\{(A^{(t)}, B^{(t)})\}$ is bounded. If $\rho > 0$, then every accumulation point $(A^*, B^*)$ of Algorithm 2 is a first-order stationary point of $\Phi$, i.e.*

$$\nabla_A \Phi(A^*, B^*) = 0, \qquad \nabla_B \Phi(A^*, B^*) = 0.$$

*Remark* 1. As $\Phi$ is a polynomial in $(A, B)$, it is real-analytic and satisfies the Kurdyka–Łojasiewicz (KŁ) property. Thus, for our proximal alternating-minimization with exact block updates and $\rho > 0$, standard KŁ results ensure that $\{(A^{(t)}, B^{(t)})\}$ converges to a stationary point with sublinear rate (Attouch & Bolte, 2009; Attouch et al., 2013; Bolte et al., 2014).

### 3.3 Uncertainty-Weighted Multi-Objective Distillation (UW-MOD)

After computing the low-rank factors $\{(A_\ell, B_\ell)\}_{\ell=1}^L$ via ALS, the compressed student is further adapted on the same calibration corpus used earlier, by aligning it with a frozen teacher. Let $(x, y) \sim \mathcal{D}_{\text{cal}}$ denote calibration inputs (with labels when available). For any $x$, denote the student's layerwise features and logits by $\{f_\ell^s(x)\}$ and $z^s(x)$, and the teacher's by $\{f_\ell^t(x)\}$ and $z^t(x)$.

To transfer knowledge effectively, we aggregate three complementary losses—(i) cross-entropy (CE), (ii) knowledge distillation (KD), and (iii) feature distillation (FD)—and weight them by learned task uncertainties to automatically calibrate heterogeneous loss scales and noise levels.

**Cross-Entropy (CE).** When labels are available, the student is trained with the standard cross-entropy loss on logits $z^s(x)$:

$$\mathcal{L}_{\text{CE}} = \mathbb{E}_{(x,y)\sim\mathcal{D}_{\text{cal}}}\big[\text{CE}\big(y,\ \text{softmax}(z^s(x))\big)\big].$$

This term is omitted when labels are unavailable.

**Knowledge Distillation (KD).** To align predictive distributions, we minimize the Kullback–Leibler (KL) divergence between temperature-scaled teacher and student logits:

$$\mathcal{L}_{\text{KD}} = T^2\, \mathbb{E}_{x\sim\mathcal{D}_{\text{cal}}}\Big[\text{KL}\Big(\text{softmax}\big(z^t(x)/T\big) \,\Big\|\, \text{softmax}\big(z^s(x)/T\big)\Big)\Big],$$

where $T > 0$ is the temperature. The prefactor $T^2$ scales the gradient across different temperatures.

**Feature Distillation (FD).** Beyond CE and KD, we align selected intermediate representations within attention and feed-forward blocks. For each chosen block $\ell \in \mathcal{S}_{\text{feat}}$, we distill a sparse set of sites including Q/K/V projections, Att, Head, the multi-head attention output (MHA), and the two feed-forward activations (FF1/FF2). Although MHA summarizes Q/K/V interactions, we include Att and Head explicitly to probe finer-grained alignment within attention. Let $u_{r,\ell}^{(\cdot)}(x)$ denote the student/teacher tensor at site $r$, and alignment is enforced by a weighted mean squared error:

$$\mathcal{L}_{\text{FD}} = \mathbb{E}_x\Big[ \sum_{\ell\in\mathcal{S}_{\text{feat}}} \sum_{r\in\mathcal{R}_\ell} \eta_r\, \|u_{r,\ell}^s(x) - u_{r,\ell}^t(x)\|_F^2 \Big],$$

where $\eta_r$ are site weights.

**Uncertainty-based weighting.** To eliminate the need for manually tuned coefficients, each objective is associated with a homoscedastic noise parameter $\sigma_i^2$ ($i \in \{\text{CE, KD, FD}\}$) that characterizes its intrinsic uncertainty. The overall training objective is

$$\mathcal{L}_{\text{total}} = \sum_{i\in\{\text{CE,KD,FD}\}} \frac{1}{\sigma_i^2}\, \mathcal{L}_i + \log\sigma_i^2. \tag{6}$$

We reparameterize $\alpha_i = \log\sigma_i^2$ and jointly optimize $\{\alpha_i\}$ together with the student's low-rank parameters $\{A_\ell, B_\ell\}$ via gradient descent. This guarantees $\sigma_i^2 > 0$ and prevents degenerate solutions. The corresponding weights $w_i = \exp(-\alpha_i) = 1/\sigma_i^2$ adapt dynamically during training, suppressing noisy or ill-conditioned objectives (large $\sigma_i^2$) while emphasizing more reliable signals.

Table 1: Comparison with SOTA baselines on Llama-3-8B under different compression ratios. Best perplexity (green) and accuracy (red) are highlighted.

| Ratio | Method | WikiText2 | PTB | C4 | Avg (PPL) | ARC_e | WinoG. | HellaS. | PIQA | Avg (Acc) |
|-------|--------|-----------|-----|-----|-----------|-------|--------|---------|------|-----------|
| 0% (16G) | Original | 8.28 | 13.01 | 13.23 | 11.51 | 76.14 | 70.00 | 46.70 | 78.60 | 67.86 |
| 40%(9.6G) | Lillama | 11512.88 | 6685.53 | 10988.46 | 9728.96 | 25.34 | 49.50 | 27.40 | 51.30 | 38.39 |
| | SVD-LLM V2 | 45.13 | 79.32 | 118.87 | 81.11 | 34.43 | 49.70 | 30.70 | 59.70 | 43.63 |
| | SoLA | 85.43 | 293.46 | 286.02 | 221.64 | 30.81 | 51.30 | 29.60 | 57.30 | 42.25 |
| | LoRKD | 20.15 | 68.12 | 74.21 | 54.16 | 58.39 | 57.50 | 37.10 | 57.40 | 52.60 |
| | ALS-ActLR | 9.48 | 37.05 | 43.57 | 30.03 | 60.23 | 60.80 | 39.20 | 69.90 | 57.53 |
| 60%(6.4G) | Lillama | 18580.33 | 13440.07 | 17969.86 | 16663.42 | 25.46 | 49.70 | 27.10 | 50.70 | 38.24 |
| | SVD-LLM V2 | 704.41 | 1860.61 | 1166.05 | 1243.69 | 25.46 | 48.30 | 28.90 | 55.00 | 39.42 |
| | SoLA | 244.86 | 712.84 | 571.15 | 509.62 | 26.94 | 50.30 | 29.10 | 56.30 | 40.66 |
| | LoRKD | 55.19 | 236.11 | 153.94 | 148.41 | 40.94 | 51.20 | 31.70 | 59.50 | 45.84 |
| | ALS-ActLR | 15.60 | 112.13 | 88.24 | 71.99 | 44.70 | 54.40 | 35.30 | 62.70 | 49.27 |
| 80%(3.2G) | Lillama | 18460.33 | 12618.15 | 15934.28 | 15670.92 | 24.79 | 49.10 | 27.50 | 52.00 | 38.35 |
| | SVD-LLM V2 | 50487.72 | 71100.31 | 49018.99 | 56869.01 | 24.54 | 48.50 | 26.80 | 52.70 | 38.14 |
| | SoLA | 420.94 | 1588.09 | 1010.00 | 1006.34 | 25.88 | 48.60 | 28.80 | 55.00 | 39.57 |
| | LoRKD | 317.97 | 1846.15 | 989.42 | 1051.18 | 29.51 | 49.20 | 29.10 | 54.90 | 40.68 |
| | ALS-ActLR | 199.37 | 1564.00 | 892.37 | 885.25 | 31.86 | 50.60 | 30.40 | 56.60 | 42.37 |

*Remark* 2. (i) *Scale calibration.* The three objectives operate on different numerical scales; Eq. (6) automatically harmonizes them through learned weights improving optimization stability and preventing gradient domination.

(ii) *Noise adaptivity.* On small calibration sets, label noise and teacher–student mismatch vary across losses; the uncertainty parameters down-weight noisier objectives, enhancing robustness.

(iii) *Computational profile.* In practice, we precompute teacher logits and selected features once on the calibration set and cache them; UW-MOD then only requires student forward passes per step, plus a small number of scalar uncertainty parameters.

## 4 EXPERIMENTS

We compare *ALS-ActLR* with four *LRA–then–update* baselines: LoRKD (Ben Noach & Goldberg, 2020), SVD-LLM V2 (Wang et al., 2025b;a), Lillama (Sy et al., 2025), and SoLA (Huang et al., 2025). Models cover (i) a range of backbones of different sizes—TinyLLaMA-1.1B, OpenLLaMA-3B, LLaMA-3-8B, LLaMA-7B and LLaMA-2-13B—and (ii) four 7B-scale families—OLMoE, Vicuna, Tulu, and WizardLM (Touvron et al., 2023; Chiang et al., 2023; Ivison et al., 2023; Xu et al., 2025). We evaluate on seven benchmarks: three language modeling corpora (WikiText-2, PTB, C4) and four commonsense reasoning tasks (ARC-e, WinoGrande, HellaSwag, PIQA) (Merity et al., 2016; Marcus et al., 1993; Raffel et al., 2019; Sakaguchi et al., 2019; Zellers et al., 2019; Bisk et al., 2019; Clark et al., 2018). Following ASVD (Yuan et al., 2024), all methods use a calibration-based setting with 256 sentences sampled from WikiText-2. Experiments run on an H20-NVLink (96 GB) platform; additional details appear in the Appendix A.1. Throughout, we define the *compression ratio* as the proportion of model parameters that are *removed*. For example, a 60% ratio indicates that 60% of the parameters are removed, retaining 40% of the original size.

### 4.1 OVERALL PERFORMANCE

Tables 1, 2, and 3 present results for Llama-3-8B, Llama-2-13B, and Llama-7B at 40%, 60%, and 80% compression, evaluated in terms of language modeling perplexity (PPL; lower is better) and commonsense reasoning accuracy (Acc; higher is better). Table 4 summarizes the same benchmarks across four 7B backbones at 60% compression. Table 5 summarizes the benchmark results at 60% compression across four backbones of different sizes, where 1.1B corresponds to TinyLLaMA-1.1B, 3B to OpenLLaMA-3B, and 7B/13B to LLaMA-7B and LLaMA-2-13B, respectively.

**Language modeling perplexity.** Across all three Llama backbones and all compression levels, *ALS-ActLR* consistently achieves the lowest average perplexity among compressed models. It improves over all four *LRA–then–update* baselines (LoRKD, SVD-LLM V2, SoLA, and Lillama) on Llama-3-8B, Llama-2-13B, and Llama-7B at 40–80% compression (Tables 1–3). In contrast, SVD-

Table 2: Comparison with SOTA baselines on Llama-2-13B under different compression ratios. Best perplexity (green) and accuracy (red) are highlighted.

| Ratio | Method | WikiText2 | PTB | C4 | Avg (PPL) | ARC_e | WinoG. | HellaS. | PIQA | Avg (Acc) |
|---|---|---|---|---|---|---|---|---|---|---|
| 0% (26 G) | Original | 6.11 | 34.62 | 8.52 | 16.42 | 74.49 | 69.10 | 49.50 | 78.20 | 67.82 |
| 40%(15.6G) | Lillama | 162.42 | 1829.82 | 368.75 | 787.00 | 53.96 | 57.30 | 37.50 | 66.70 | 53.86 |
| | SVD-LLM V2 | 9.84 | 125.47 | 22.01 | 52.44 | 58.33 | 60.10 | 40.30 | 66.20 | 56.23 |
| | SoLA | 15.64 | 2274.12 | 40.04 | 776.60 | 44.19 | 55.60 | 35.20 | 60.60 | 48.90 |
| | LoRKD | 13.98 | 234.19 | 29.44 | 92.54 | 60.98 | 58.80 | 40.80 | 67.10 | 56.92 |
| | ALS-ActLR | 6.96 | 69.44 | 14.44 | 30.28 | 68.06 | 63.10 | 44.90 | 72.50 | 62.14 |
| 60%(10.4G) | Lillama | 15870.41 | 4631.88 | 35591.35 | 18697.88 | 25.55 | 50.20 | 28.00 | 51.00 | 38.69 |
| | SVD-LLM V2 | 20.27 | 413.29 | 74.92 | 169.49 | 36.32 | 50.00 | 34.10 | 55.80 | 44.05 |
| | SoLA | 40.22 | 6894.60 | 116.84 | 2350.55 | 28.03 | 48.50 | 30.20 | 53.80 | 40.13 |
| | LoRKD | 24.61 | 381.14 | 68.16 | 157.97 | 52.84 | 53.80 | 37.60 | 62.40 | 51.66 |
| | ALS-ActLR | 9.18 | 155.36 | 22.17 | 62.24 | 57.66 | 58.20 | 40.70 | 65.00 | 55.39 |
| 80%(5.2G) | Lillama | 79634.08 | 6643.43 | 310129.87 | 132135.79 | 25.13 | 49.20 | 27.90 | 51.20 | 38.36 |
| | SVD-LLM V2 | 239.14 | 1989.80 | 831.99 | 1020.31 | 26.01 | 50.70 | 28.30 | 53.00 | 39.50 |
| | SoLA | 142.79 | 7563.90 | 367.70 | 2691.46 | 25.46 | 49.10 | 28.90 | 52.60 | 39.02 |
| | LoRKD | 68.48 | 2843.87 | 134.55 | 1015.63 | 32.48 | 49.40 | 31.80 | 54.20 | 41.97 |
| | ALS-ActLR | 18.20 | 1301.42 | 70.11 | 463.24 | 36.15 | 51.50 | 33.70 | 55.60 | 44.24 |

Table 3: Comparison with SOTA baselines on Llama-7B under different compression ratios. Best perplexity (green) and accuracy (red) are highlighted.

| Ratio | Method | WikiText2 | PTB | C4 | Avg (PPL) | ARC_e | WinoG. | HellaS. | PIQA | Avg (Acc) |
|---|---|---|---|---|---|---|---|---|---|---|
| 0% (13G) | Original | 5.68 | 8.35 | 7.34 | 7.12 | 71.25 | 68.19 | 42.10 | 77.90 | 64.86 |
| 40% (7.8G) | Lillama | 441744.93 | 424367.59 | 400819.90 | 422310.80 | 23.55 | 49.80 | 26.00 | 50.50 | 37.46 |
| | SVD-LLM V2 | 13.49 | 56.72 | 49.32 | 39.84 | 28.46 | 49.81 | 31.52 | 57.20 | 41.75 |
| | SoLA | 26.67 | 261.71 | 63.41 | 117.26 | 29.88 | 49.20 | 30.20 | 56.40 | 41.42 |
| | LoRKD | 10.61 | 24.58 | 18.98 | 18.06 | 55.89 | 53.83 | 37.90 | 66.60 | 53.56 |
| | ALS-ActLR | 8.45 | 22.91 | 17.56 | 16.31 | 56.61 | 57.14 | 39.20 | 67.00 | 54.99 |
| 60% (5.2G) | Lillama | 483574.12 | 380813.81 | 404593.48 | 422993.80 | 24.83 | 50.50 | 26.30 | 50.40 | 38.01 |
| | SVD-LLM V2 | 54.82 | 381.41 | 331.45 | 255.89 | 25.16 | 48.71 | 29.10 | 54.80 | 39.44 |
| | SoLA | 69.34 | 577.51 | 172.13 | 272.99 | 25.42 | 49.60 | 28.50 | 54.90 | 39.61 |
| | LoRKD | 27.73 | 169.79 | 70.87 | 89.46 | 37.21 | 50.20 | 33.50 | 59.90 | 45.20 |
| | ALS-ActLR | 10.34 | 43.48 | 29.39 | 27.74 | 44.49 | 54.30 | 35.00 | 61.90 | 48.92 |
| 80% (2.6G) | Lillama | 277712.53 | 283608.16 | 323623.13 | 294981.27 | 22.61 | 51.10 | 26.10 | 51.90 | 37.93 |
| | SVD-LLM V2 | 755.48 | 5351.70 | 2924.10 | 3010.43 | 24.90 | 50.50 | 27.70 | 52.30 | 38.85 |
| | SoLA | 200.64 | 1017.34 | 406.92 | 541.63 | 26.01 | 49.50 | 28.30 | 52.30 | 39.03 |
| | LoRKD | 61.09 | 770.63 | 263.86 | 365.19 | 28.49 | 50.91 | 29.50 | 53.10 | 40.50 |
| | ALS-ActLR | 28.30 | 156.94 | 97.15 | 94.13 | 32.03 | 51.54 | 32.50 | 56.00 | 43.02 |

LLM V2, SoLA, and Lillama often exhibit sharp perplexity degradation under aggressive compression, and in some cases become numerically unstable when only a small calibration set is available. The aggregated results at 60% compression further support this trend: *ALS-ActLR* attains the best average perplexity not only across the four 7B families (OLMoE, Vicuna, Tulu, WizardLM; Table 4) but also across backbones of different sizes from TinyLLaMA-1.1B to LLaMA-2-13B (Table 5). These patterns indicate that *ALS-ActLR* produces more faithful low-rank approximations and leads to a smoother, more stable perplexity profile as compression increases.

**Commonsense reasoning accuracy.** *ALS-ActLR* also brings consistent gains on downstream commonsense benchmarks. On all three Llama backbones, it matches or surpasses the best-performing baseline at every compression ratio, with the most noticeable advantages emerging at moderate and high compression where other methods tend to suffer larger drops in accuracy (Tables 1–3). Relative to SVD-LLM V2, SoLA, and Lillama, *ALS-ActLR* maintains substantially higher accuracies, showing that improved calibration of the predictive distribution is not limited to perplexity but transfers to the quality of discrete decisions. The averaged results at 60% compression confirm that *ALS-ActLR* is the best-performing method across all four 7B families (Table 4) and across the 1.1B, 3B, 7B, and 13B scales (Table 5). Overall, the method preserves more of the original model's reasoning ability under tight parameter budgets.

Table 4: Comparison with SOTA baselines across four 7B backbones at 60% compression. Best perplexity (green) and accuracy (red) are highlighted.

| | Average PPL | | | | | Average Accuracy (%) | | | |
|---|---|---|---|---|---|---|---|---|---|
| Method | OLMoE | Vicuna | Tulu | WizardLM | Method | OLMoE | Vicuna | Tulu | WizardLM |
| Lillama | 814.74 | 150598.17 | 148560.72 | 199421.11 | Lillama | 40.46 | 37.75 | 37.46 | 38.24 |
| SVD-LLM V2 | 32.81 | 612.86 | 982.41 | 592.33 | SVD-LLM V2 | 51.90 | 40.11 | 39.51 | 41.67 |
| SoLA | 80.19 | 4889.06 | 6243.71 | 280.85 | SoLA | 48.81 | 38.91 | 38.41 | 40.70 |
| LoRKD | 38.77 | 393.33 | 149.00 | 160.36 | LoRKD | 50.16 | 40.03 | 45.68 | 43.10 |
| ALS-ActLR | 25.53 | 111.79 | 144.70 | 66.51 | ALS-ActLR | 55.04 | 47.14 | 46.44 | 48.52 |

Table 5: Comparison with SOTA baselines across backbones of different sizes at 60% compression. Best perplexity (green) and accuracy (red) are highlighted.

| | Average PPL | | | | | Average Accuracy (%) | | | |
|---|---|---|---|---|---|---|---|---|---|
| Method | 1.1B | 3B | 7B | 13B | Method | 1.1B | 3B | 7B | 13B |
| Lillama | 62383.88 | 57673.69 | 422993.80 | 18697.88 | Lillama | 37.38 | 39.15 | 38.01 | 38.69 |
| SVD-LLM V2 | 468.51 | 349.16 | 255.89 | 169.49 | SVD-LLM V2 | 39.74 | 40.19 | 39.44 | 44.05 |
| SoLA | 249.52 | 354.71 | 272.99 | 2350.55 | SoLA | 40.15 | 39.46 | 39.61 | 40.13 |
| LoRKD | 477.13 | 378.61 | 89.46 | 157.97 | LoRKD | 37.36 | 38.27 | 45.20 | 51.66 |
| ALS-ActLR | 132.62 | 318.74 | 27.74 | 62.24 | ALS-ActLR | 41.52 | 40.78 | 48.92 | 55.39 |

**Cross-model generalization and stability.** All methods inevitably degrade under extreme compression, but they exhibit distinct stability profiles. Lillama is highly sensitive to the limited calibration set and can become severely unstable at higher compression ratios. SVD-LLM V2 behaves more reliably at moderate ratios, yet its performance drops sharply under very aggressive compression. SoLA avoids catastrophic failures but consistently lags behind the other approaches. LoRKD is relatively robust compared to the other baselines, yet *ALS-ActLR* still consistently outperforms it across both perplexity and accuracy. Overall, *ALS-ActLR* shows no catastrophic breakdowns and maintains a clear advantage across Llama-3-8B, Llama-2-13B, Llama-7B, the four 7B families including OLMoE, and all considered parameter scales (Tables 1–5). This consistent ranking across architectures and sizes demonstrates that the proposed *ALS-ActLR* generalizes well and yields a stable, predictable trade-off between compression and performance.

Table 6: Average PPL and Accuracy comparison for methods at 60% compression on Llama-7B.

| Method | PPL | Accuracy (%) |
|---|---|---|
| SVD | 97733.48 | 37.94 |
| TADW | 376.26 | 39.83 |
| **ALS** | **364.15** | **40.02** |

Table 7: Layer-wise reconstruction error of Llama-7B at 60% compression with the $Q$ projection matrix, comparing SVD, TADW, and Activation-aware ALS.

| Method | Layer 1 | Layer 6 | Layer 11 | Layer 16 | Layer 21 | Layer 26 | Layer 31 |
|---|---|---|---|---|---|---|---|
| SVD | 22.95 | 706.81 | 732.03 | 856.19 | 1028.59 | 1233.48 | 1110.17 |
| TADW | 85.27 | 496.39 | 556.72 | 675.35 | 789.50 | 955.53 | 879.97 |
| **ALS** | **10.33** | **488.21** | **546.06** | **666.74** | **783.90** | **951.99** | **876.16** |

## 4.2 ABLATION STUDIES

**Effectiveness of Activation-aware ALS.** We compare activation-aware ALS against SVD (the unparameterized distillation in LoRKD) and truncation-aware data whitening (TADW; the unparameterized update in SVD-LLM) on Llama-7B at 60% compression. As shown in Table 6, ALS achieves a substantially lower average PPL (364.15) than SVD (97733.48) and TADW (376.26), while also yielding the highest accuracy (40.02%). Reconstruction errors $\|WS - ABS\|_F$ on selected $Q$-projection layers (Table 7) are consistently smallest for ALS, confirming tighter activation-weighted approximations across depth. Table 9 further reports layer-wise reconstruction errors for all projection matrices, where ALS again attains the lowest errors across most layers. These results confirm that Activation-aware ALS provides a more efficient compression solution, achieving better PPL, accuracy, and reconstruction error compared to traditional methods, thereby highlighting its effectiveness in preserving activation fidelity and improving model compression.

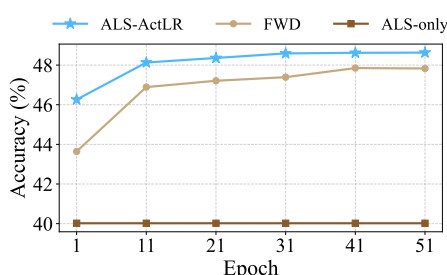

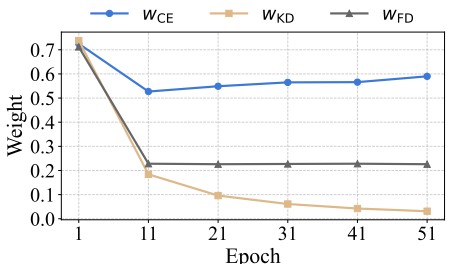

Figure 3: Average accuracy progression of ALS-only, FWD, and ALS-ActLR over training epochs at 60% compression on Llama-7B.

Figure 4: Learned uncertainty weight trajectories in ALS-ActLR on Llama-7B with 60% compression.

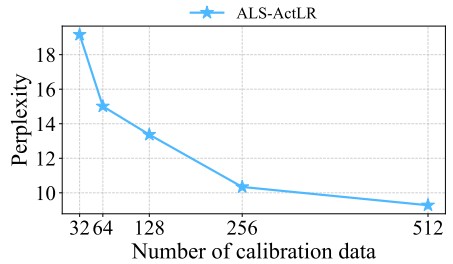

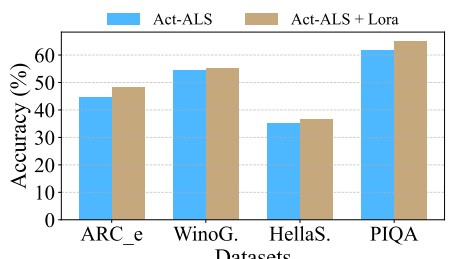

Figure 5: Perplexity on WikiText-2 vs. calibration set size for ALS-ActLR on Llama-7B at 60% compression.

Figure 6: Accuracy of Act-ALS on LLaMA-7B at 60% compression with and without LoRA fine-tuning.

**Contribution of uncertainty-weighted distillation.** We evaluate three variants of the same ALS-compressed student: *ALS-only* (without distillation), *fixed-weight distillation* (FWD), and *uncertainty-weighted distillation* (the full *ALS-ActLR*). On Llama-7B under 60% compression, Figure 3 indicates that uncertainty weighting achieves both higher and more stable accuracy during training, reaching 48.63% compared to 47.83% for FWD and 40.02% for ALS-only. The learned weights (Figure 4) adaptively rebalance objectives: CE remains dominant, KD decays quickly, and FD stabilizes at a moderate level. The down-weighting of KD reflects the diminishing value of soft targets once the student has absorbed the teacher's coarse distributional patterns, particularly under limited calibration data where teacher signals may be noisy. By contrast, CE is tied directly to ground-truth labels and thus provides the most reliable supervision, explaining its consistently high weight. FD occupies an intermediate role, offering structural alignment without overwhelming optimization. Together, these dynamics prevent any single objective from dominating and enable the student to exploit complementary signals more effectively.

## 4.3 CALIBRATION SET SIZE AND PERPLEXITY

We vary the calibration set size at 60% compression on Llama-7B and evaluate perplexity on the WikiText-2 test set. Figure 5 shows monotonically decreasing PPL with diminishing returns beyond about 128 sentences, plateauing around 256–512 samples. Thus, *ALS-ActLR* is effective under tight calibration budgets yet continues to benefit from modestly larger sets.

To investigate the effect of different calibration sets, we further evaluate the Act-ALS (the ALS compression stage of ALS-ActLR) compressed models on both PPL and accuracy when using WikiText-2 or C4 as the calibration data. Here, "WikiText-2" and "C4" in the figures denote the type of calibration set used in the compression stage. As shown in Figures 7 and 8, each calibration set naturally performs better on its own evaluation benchmark, since the compressed model is more familiar with the distribution it has been calibrated on. Apart from this expected advantage on

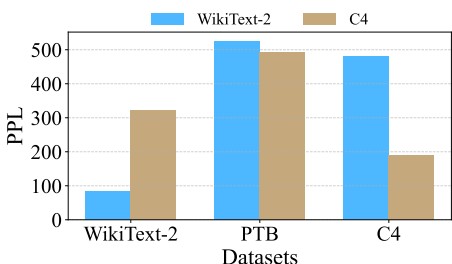 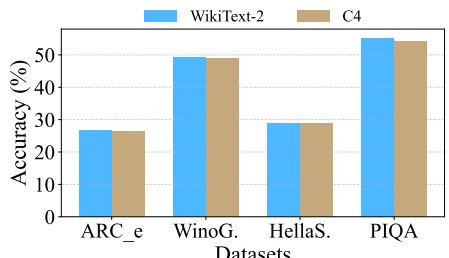

Figure 7: Perplexity comparison of Act-ALS compressed models when using WikiText-2 or C4 as calibration sets.

Figure 8: Accuracy comparison of Act-ALS compressed models when using WikiText-2 or C4 as calibration sets.

the matching dataset, the performances across other benchmarks are very close, indicating that our method is largely insensitive to the choice of calibration set.

### 4.4 PERFORMANCE WITH LoRA FINE-TUNING

Low-Rank Adaptation (LoRA) is a widely used parameter-efficient fine-tuning technique that inserts trainable low-rank matrices into a frozen backbone model, allowing efficient adaptation with only a small number of additional parameters. Unlike full fine-tuning, LoRA updates only a lightweight set of parameters, which is especially beneficial when combined with compressed models.

To assess the complementary benefits of LoRA, we applied it on top of *Act-ALS*-compressed LLaMA-7B (60% compression). Figure 6 shows pre-/post-LoRA accuracy on four common-sense benchmarks (ARC_e, WinoGrande, HellaSwag, PIQA): LoRA yields consistent gains of +1.0%–+3.9% on all tasks, with especially strong improvements on ARC_e and PIQA and no degradation elsewhere, indicating that it helps the compressed student recover knowledge-intensive reasoning. Overall, LoRA provides a lightweight yet effective enhancement to *ALS-ActLR* via task-specific adaptation. Experimental details are given in Appendix A.5.

### 4.5 INFERENCE EFFICIENCY UNDER VARYING COMPRESSION RATIOS

Figure 13 reports the inference performance of models compressed by *ALS-ActLR*, including throughput, time-to-first-token (TTFT), per-token latency, and peak memory across batch sizes for 40%, 60%, and 80% compression. As shown in the figure, higher compression consistently improves runtime efficiency: throughput scales more favorably at larger batches; TTFT and per-token latency decline with stronger compression; and peak memory drops substantially, enabling larger batches under the same hardware constraints. These trends underscore the deployability of *ALS-ActLR* in resource-limited settings. As shown in Table 3, however, very high compression also entails performance degradation. Thus, selecting an appropriate compression ratio enables practitioners to balance accuracy and efficiency according to application requirements.

## 5 CONCLUSION

We presented *ALS-ActLR*, an activation-aware low-rank compression framework within the *LRA–then–update* paradigm that integrates (i) SIMT for reparameterizing activation-weighted errors, (ii) alternating least-squares with ridge and proximal regularization to *directly* minimize $\|WX - ABX\|_F^2$, and (iii) uncertainty-weighted multi-objective distillation to adaptively balance CE, KD, and feature alignment. We establish convex block subproblems, sufficient descent, and convergence to stationary points under the KŁ property. Empirically, *ALS-ActLR* outperforms strong baselines across multiple model families and scales on seven benchmarks under 40–80% compression, and higher compression further reduces latency, memory usage, and inference cost. Overall, *ALS-ActLR* demonstrates that activation-aware compression can be both theoretically principled and practically effective for scaling down large models. Looking ahead, it offers a foundation for integrating with emerging techniques to further advance efficient on-device deployment of LLMs.

## REPRODUCIBILITY STATEMENT

An anonymized implementation of *ALS-ActLR*, including full algorithmic details and settings, is provided in the Supplementary Material, enabling reproduction of the method and its reported results. This ensures transparency and facilitates future research built upon our approach.

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

| Dataset | Task Type | Train | Dev | Test |
|---------|-----------|-------|-----|------|
| WikiText2 | Language Modeling | 36718 | 3760 | 4358 |
| PTB | Language Modeling | 42068 | 3370 | 3761 |
| C4 | Language Modeling | - | 45576 | - |
| ARC-e | Commonsense/QA | 2251 | 570 | 2376 |
| WinoGrande | Commonsense/QA | 40398 | 1267 | 1767 |
| HellaSwag | Commonsense/QA | 39905 | 10042 | – |
| PIQA | Commonsense/QA | 16113 | 1838 | 3084 |

Table 8: Datasets and splits used in our evaluation (counts reflect local cached corpora).

# A  APPENDIX

## A.1  EXPERIMENTAL SETUP

**Hardware and software.**  All experiments are conducted on a single NVIDIA H20-NVLink GPU (96 GB). We use PyTorch 2.1.2 with CUDA 12.1 and cuDNN 8.9.2, with TF32 acceleration enabled. Forward passes run in `bfloat16`, while statistics accumulation and ALS updates are performed in `float32`. The random seed is fixed to 3 unless otherwise stated.

**Models and datasets.**  We evaluate four 7B-scale backbones from HuggingFace: LLaMA-7B (jeffwan/llama-7b-hf), Tulu-7B (allenai/tulu-2-7b), Vicuna-7B (lmsys/vicuna-7b-v1.5), and WizardLM-7B (TheBloke/wizardLM-7B-HF). Language modeling is evaluated on WikiText-2, PTB, and C4, while commonsense reasoning is measured on ARC-e, WinoGrande, HellaSwag, and PIQA. Dataset statistics and splits are summarized in Table 8.

**Calibration and SIMT statistics.**  We construct a tiny calibration set of 256 sentences sampled from WikiText-2 with no overlap with evaluation data. Second-moment matrices $M_\ell$ are accumulated in a streaming manner over tokens to avoid storing full activations: $M_\ell \leftarrow \sum_b X_{\ell,b} X_{\ell,b}^\top$. For each layer, we compute the spectral factor $S_\ell$ via eigendecomposition to obtain $S_\ell S_\ell^\top = M_\ell$ (SIMT reparameterization).

**Rank Allocation**  To enforce a global compression ratio while ensuring fairness across baselines, we allocate layer ranks $\{r_\ell\}$ proportionally to the parameter counts of the original weight matrices. Concretely, for a linear weight $W \in \mathbb{R}^{m \times n}$, the target rank $r$ is computed as

$$r = \texttt{int}\left(\frac{m \times n \times (1 - \text{ratio})}{m + n}\right),$$

where `ratio` denotes the proportion of parameters removed (e.g., 0.6 corresponds to 60% compression, i.e., retaining 40%). This rule balances the contribution of each layer to the global budget and avoids manual tuning. All methods compared in the main text use the same set of ranks under a given global ratio.

**Compression protocol and rank allocation.**  We target global compression ratios of 40%, 60%, and 80% (defined as the proportion of parameters *removed*). Layer ranks $\{r_\ell\}$ are allocated proportionally to the original layer parameter counts under the global budget and held fixed across baselines for fair comparison. ALS is initialized from the top-$r_\ell$ SVD of $W_\ell$.

**ALS solver.**  For each layer, we solve the activation-aware objective

$$\tfrac{1}{2}\|W S_\ell - A_\ell B_\ell S_\ell\|_F^2 \;+\; \tfrac{\tau}{2}\left(\|A_\ell\|_F^2 + \|B_\ell\|_F^2\right)$$

via alternating least squares (ALS). Here $\tau$ (default 0.003) is the ridge penalty that improves conditioning and generalization. At each iteration, we further stabilize updates with proximal terms $\tfrac{\rho}{2}\|A_\ell - A_\ell^{(t)}\|_F^2$ and $\tfrac{\rho}{2}\|B_\ell - B_\ell^{(t)}\|_F^2$, where $\rho$ (default 0.003) controls the step size of the algorithm. Both subproblems remain convex quadratics and admit efficient closed-form solutions. The $A$-step solves an $r_\ell \times r_\ell$ linear system via Cholesky; the $B$-step is diagonalized by basis rotations and reduces to elementwise updates. We use a fixed number of ALS iterations $T_{\text{ALS}}$ (default 3).

**Uncertainty-weighted distillation.** After ALS, the student is adapted to a frozen teacher on the calibration set by optimizing $\mathcal{L}_{\text{total}} = \sum_{i \in \{\text{CE,KD,FD}\}} e^{-\alpha_i} \mathcal{L}_i + \alpha_i$ with $\alpha_i = \log \sigma_i^2$. We update low-rank factors with learning rate $\eta_1$ (default: $1 \times 10^{-5}$) and the log-uncertainties with $\eta_2$ (default 0.01) for $T_{\text{dist}}$ epochs (default 50). Temperature for KD is denoted by $T$ (default 2.0).

**Feature sites and stride.** We align a sparse set of transformer blocks with stride $s$ (default 4). Candidate feature sites per aligned block include $\{Q, K, V, \text{Head}, \text{Att}, \text{MHA}, \text{FF1}, \text{FF2}\}$. Empirically, moderate strides ($s = 3 \sim 4$) balance knowledge transfer and cost. Unless otherwise specified, site weights are set as Q/K/V=1, Att=0.25, Head=0.5, MHA=1, FF1=0.5, and FF2=1.

**Evaluation protocol.** For language modeling, we report perplexity (lower is better) on the test splits of WikiText-2, PTB, and C4. For commonsense reasoning, we report accuracy (higher is better) on ARC-e, WinoGrande, HellaSwag, and PIQA with the official evaluation scripts. All baselines share the same calibration set, rank allocation, and evaluation pipeline.

**Inference efficiency.** Throughput (tok/s), time-to-first-token (TTFT), per-token latency, and peak memory are measured across batch sizes on the same hardware. We use greedy decoding for timing unless otherwise noted. Each configuration is repeated multiple times and we report the mean.

## A.2 ALS-ACTLR: FULL PROCEDURE

For completeness, we summarize the entire *ALS-ActLR* pipeline—combining spectral-informed metric transformation, activation-aware ALS, and uncertainty-weighted distillation—as the following Algorithm 1.

## A.3 DETAILED DERIVATIONS OF ACTIVATION-AWARE ALS

Building on (2), we adopt alternating least squares (ALS) with ridge regularization and a proximal term. The two block updates take the quadratic forms given in the main text. We now provide their detailed solutions.

**Update for $A$.** With $B = B^{(t)}$, set

$$H := BS \in \mathbb{R}^{r \times n}, \qquad G_A := HH^\top + (\tau + \rho)I_r, \qquad R_A := HM^\top + \rho A^{(t)\top}.$$

The normal equations are

$$A^{(t+1)} G_A = MH^\top + \rho A^{(t)}.$$

Rather than forming $G_A^{-1}$, factor $G_A = LL^\top$ (Cholesky) and solve

$$LY = R_A, \qquad L^\top X = Y, \qquad A^{(t+1)} \leftarrow X^\top.$$

With $\tau + \rho > 0$, $G_A \succ 0$ yields a unique, stable update; if $\tau = \rho = 0$, uniqueness requires $\text{rank}(H) = r$. This design reduces computation to an $r \times r$ system, while the positive definiteness of $G_A$ ensures well-conditioned, stable updates across iterations.

**Update for $B$.** With $A = A^{(t+1)}$, the normal equations read

$$(A^\top A) B (SS^\top) + (\tau + \rho) B = A^\top MS^\top + \rho B^{(t)}. \tag{7}$$

Let $A = U_A \Lambda_A V_A^\top$ and $S = U_S \Lambda_S V_S^\top$ denote their SVDs. Rotate to diagonal bases via

$$B' := V_A^\top B U_S, \qquad C := \Lambda_A U_A^\top M V_S \Lambda_S + \rho V_A^\top B^{(t)} U_S.$$

Then (7) decouples elementwise:

$$B'(j, \ell) = \frac{C(j, \ell)}{\Lambda_A^2(j, j) \Lambda_S^2(\ell, \ell) + \tau + \rho} \quad \forall j \in [r], \ell \in [n], \tag{8}$$

and we map back via $B^{(t+1)} \leftarrow V_A B' U_S^\top$. This diagonalization reduces the update to elementwise divisions, yielding both computational efficiency and numerically stable solutions.

---

**Algorithm 1** ALS-ActLR

---

**Require:** Teacher model with linear weights $\{W_\ell\}_{\ell=1}^L$; small calibration set $\mathcal{D}_{\text{cal}}$; target ranks $\{r_\ell\}$; regularization $\tau \geq 0$, proximal $\rho \geq 0$; ALS iters $T_{\text{ALS}}$; distillation epochs $T_{\text{dist}}$; temperature $T$; feature sites $\mathcal{R}_\ell$; stride $s$ (align every $s$-th block); learning rates $\eta_1, \eta_2$.

**Ensure:** Compressed student with low-rank factors $\{A_\ell, B_\ell\}$.

 1: **(Calibration & statistics)**
 2: Initialize *student* from *teacher* weights $\{W_\ell\}$; freeze *teacher*.
 3: Run *teacher* on the calibration set $\mathcal{D}_{\text{cal}}$ to obtain per-layer inputs $X_\ell(x)$ to the linear maps $W_\ell$.
 4: Accumulate second-moment matrices by streaming: $M_\ell \leftarrow \frac{1}{|\mathcal{D}_{\text{cal}}|} \sum_{x \in \mathcal{D}_{\text{cal}}} X_\ell(x)^\top X_\ell(x)$.
 5: **(SIMT: spectral-informed metric transformation)**
 6: **for** each layer $\ell = 1, \ldots, L$ **do**
 7:    Compute spectral factor $S_\ell$ from $M_\ell$ (eigendecomposition), ensuring $S_\ell S_\ell^\top = M_\ell$.
 8: **end for**
 9: **(Activation-aware ALS per layer)**
10: **for** layer $\ell = 1, \ldots, L$ **do**
11:    **Call** Algorithm 2 with $(W_\ell, S_\ell, r_\ell, \tau, \rho, T_{\text{ALS}})$ to obtain $(A_\ell, B_\ell)$.
12:    Replace $W_\ell \leftarrow A_\ell B_\ell$ in the *student*.
13: **end for**
14: **(Uncertainty-weighted multi-objective distillation)**
15: Initialize log-uncertainties $\alpha_{\text{CE}}, \alpha_{\text{KD}}, \alpha_{\text{FD}}$ (e.g., 0).
16: **// Precompute teacher signals on the calibration set**
17: **for** mini-batches $x$ from $\mathcal{D}_{\text{cal}}$ **do**
18:    Teacher forward: logits $z^t(x)$; selected features $\{u_{r,\ell}^t(x)\}$ for $\ell \in \{1, s+1, 2s+1, \ldots\}$ and $r \in \mathcal{R}_\ell$.
19:    Cache $(z^t(x), \{u_{r,\ell}^t(x)\})$.
20: **end for**
21: **// UW-MOD using cached teacher signals**
22: **for** $e = 1, \ldots, T_{\text{dist}}$ **do**
23:    **for** mini-batches $(x, y)$ from $\mathcal{D}_{\text{cal}}$ **do**
24:       Load cached teacher logits $z^t(x)$ and features $\{u_{r,\ell}^t(x)\}$.
25:       Student forward: logits $z^s(x)$; selected features $\{u_{r,\ell}^s(x)\}$ (only low-rank factors $\{A_\ell, B_\ell\}$ and the uncertainty scalars are trainable).
26:       Compute losses:

$$\mathcal{L}_{\text{CE}} = \text{CE}\big(y, \text{softmax}(z^s)\big) \quad \text{(if labels available)},$$
$$\mathcal{L}_{\text{KD}} = T^2 \, \text{KL}\big(\text{softmax}(z^t/T) \,\|\, \text{softmax}(z^s/T)\big),$$
$$\mathcal{L}_{\text{FD}} = \sum_\ell \sum_{r \in \mathcal{R}_\ell} \eta_r \, \|u_{r,\ell}^s - u_{r,\ell}^t\|_F^2.$$

27:       Reweight by uncertainties (homoscedastic noise):

$$\mathcal{L}_{\text{total}} = \sum_{i \in \{\text{CE}, \text{KD}, \text{FD}\}} e^{-\alpha_i} \, \mathcal{L}_i + \alpha_i \quad (\alpha_i = \log \sigma_i^2).$$

28:       Gradient step: update $\{A_\ell, B_\ell\}$ with lr $\eta_1$; update $\{\alpha_i\}$ with lr $\eta_2$.
29:    **end for**
30: **end for**
31: **return** compressed student $\{A_\ell, B_\ell\}$.

---

---

**Algorithm 2** ALS for Activation-Aware Low-Rank Approximation

---

**Require:** $W \in \mathbb{R}^{m \times n}$, spectral factor $S \in \mathbb{R}^{n \times n}$, rank $r$, regularization parameter $\tau$, proximal term parameter $\rho$, iterations $T = max_{iter}$.
**Ensure:** Compressed factors $A \in \mathbb{R}^{m \times r}$, $B \in \mathbb{R}^{r \times n}$.
 1: compute SVD: $W = U\Sigma V^\top$; take top-$r$ triplet $(U_r, \Sigma_r, V_r)$
 2: $A \leftarrow U_r \Sigma_r^{1/2}$, $B \leftarrow \Sigma_r^{1/2} V_r^\top$
 3: $M \leftarrow WS$ (target)
 4: **for** $t = 1, \dots, T$ **do**
 5:    **Update** $A$:
 6:    $H \leftarrow BS$
 7:    $LL^\top \leftarrow \text{Cholesky}\big(HH^\top + (\tau + \rho)I_r\big)$
 8:    $R \leftarrow HM^\top + \rho A^\top$
 9:    solve $LY = R$ (forward) and $L^\top X = Y$ (backward)
10:    $A \leftarrow X^\top$
11:    **Update** $B$:
12:    compute SVDs: $S = U_S \Lambda_S V_S^\top$, $A = U_A \Lambda_A V_A^\top$
13:    $C \leftarrow \Lambda_A U_A^\top M V_S \Lambda_S + \rho V_A^\top B U_S$
14:    *for all* $j \in [r], \ell \in [n]$:    $B'(j, \ell) \leftarrow \dfrac{C(j, \ell)}{\Lambda_A^2(j, j)\, \Lambda_S^2(\ell, \ell) + \tau + \rho}$
15:    $B \leftarrow V_A\, B'\, U_S^\top$
16: **end for**
17: **return** $(A, B)$

---

**Algorithm.** For completeness, the full pseudocode of ALS for activation-aware low-rank approximation is given below.

### A.4 PROOFS OF THEORETICAL RESULTS

**Proof of Theorem 1**

*Proof.* Consider the $A$-update. The objective can be written as

$$Q_A(A) = \tfrac{1}{2}\|M - AH\|_F^2 + \tfrac{\tau}{2}\|A\|_F^2 + \tfrac{\rho}{2}\|A - A^{(t)}\|_F^2, \qquad M := WS,\; H := BS.$$

Differentiating yields

$$\nabla_A Q_A = A(HH^\top) - MH^\top + \tau A + \rho(A - A^{(t)}).$$

Upon vectorization, using $\text{vec}(AXB) = (B^\top \otimes A)\text{vec}(X)$, the Hessian becomes

$$\nabla_A^2 Q_A = (HH^\top) \otimes I_m + (\tau + \rho)I_{mr}.$$

Since $HH^\top \succeq 0$, this is positive semidefinite; if $\tau + \rho > 0$ it is strictly positive definite, implying strong convexity and uniqueness. When $\tau = \rho = 0$, uniqueness reduces to $\text{rank}(H) = r$.

For the $B$-update, the objective is

$$Q_B(B) = \tfrac{1}{2}\|M - ABS\|_F^2 + \tfrac{\tau}{2}\|B\|_F^2 + \tfrac{\rho}{2}\|B - B^{(t)}\|_F^2,$$

and the gradient reads

$$\nabla_B Q_B = (A^\top A)B(SS^\top) - A^\top MS^\top + \tau B + \rho(B - B^{(t)}).$$

Thus

$$\nabla_B^2 Q_B = (SS^\top) \otimes (A^\top A) + (\tau + \rho)I_{rn}.$$

The same reasoning applies: positive semidefinite always, positive definite if $\tau + \rho > 0$, and uniqueness in the unregularized case requires $A^\top A \succ 0$ and $SS^\top \succ 0$. $\qquad\square$

**Proof of Theorem 2**

*Proof.* By optimality of $A^{(t+1)} = \arg\min_A Q_A(A|A^{(t)}, B^{(t)})$ we have

$$Q_A(A^{(t+1)}|A^{(t)}, B^{(t)}) \leq Q_A(A^{(t)}|A^{(t)}, B^{(t)}).$$

Unfolding the definition of $Q_A$ gives

$$\Phi(A^{(t+1)}, B^{(t)}) + \frac{\rho}{2}\|A^{(t+1)} - A^{(t)}\|_F^2 \leq \Phi(A^{(t)}, B^{(t)}),$$

which is the first inequality.

An identical argument for $B^{(t+1)} = \arg\min_B Q_B(B|A^{(t+1)}, B^{(t)})$ gives the second. Thus $\Phi$ decreases monotonically. Since $\Phi \geq 0$, it converges. If $\rho > 0$, summing the inequalities over $t$ yields

$$\sum_{t=0}^{\infty}\|A^{(t+1)} - A^{(t)}\|_F^2 < \infty, \qquad \sum_{t=0}^{\infty}\|B^{(t+1)} - B^{(t)}\|_F^2 < \infty,$$

which proves square summability of the increments. $\square$

**Proof of Theorem 3**

*Proof.* By Theorem 2, $\Phi(A^{(t)}, B^{(t)})$ converges. Under coercivity ($\tau > 0$) or boundedness, the sequence admits accumulation points. The optimality conditions of the proximal subproblems state that

$$\nabla_A\Phi(A^{(t+1)}, B^{(t)}) + \rho(A^{(t+1)} - A^{(t)}) = 0,$$

$$\nabla_B\Phi(A^{(t+1)}, B^{(t+1)}) + \rho(B^{(t+1)} - B^{(t)}) = 0.$$

From Theorem 2, the increments vanish as $t \to \infty$. Let $t_k \to \infty$ be a subsequence with $(A^{(t_k)}, B^{(t_k)}) \to (A^*, B^*)$. Continuity of $\nabla\Phi$ then implies

$$\nabla_A\Phi(A^*, B^*) = 0, \qquad \nabla_B\Phi(A^*, B^*) = 0,$$

so $(A^*, B^*)$ is stationary. $\square$

### A.5 LoRA Configuration and Training Protocol

For reproducibility, we provide the exact configuration used for the LoRA fine-tuning experiments in Section 4.4.

**Implementation.** We adopt the PEFT implementation of LoRA, following the official Alpaca-LoRA repository. Adapters are inserted into the attention projections (Q/K/V/O) and feed-forward projections (gate, down, up). The compressed model is first prepared for int8 training, after which LoRA adapters are applied.

**Hyperparameters.** We use rank $r = 8$, scaling $\alpha = 16$, dropout 0.05, and disable bias terms. Training is performed for 2 epochs with AdamW optimizer, learning rate $1 \times 10^{-4}$, global batch size 128 (micro-batch size 4), gradient accumulation steps 32, cutoff length 256, and warmup 100 steps. Validation set size is 2000. Training is run in `fp16`, with int8-prepared backbones.

**Datasets.** We use `yahma/alpaca-cleaned` for training, and hold out 2000 samples for validation.

### A.6 Reconstruction errors for all projection matrices

Table 9 reports layer-wise reconstruction errors for all projection matrices, where ALS again attains the lowest errors across most layers.

Table 9: Layer-wise reconstruction error of Llama-7B at 60% compression for all projection matrices (Q, K, V, O, Gate, Up, Down), comparing SVD, TADW, and Activation-aware ALS.

| Projection matrix | Method | Layer 1 | Layer 6 | Layer 11 | Layer 16 | Layer 21 | Layer 26 | Layer 31 |
|---|---|---|---|---|---|---|---|---|
| Q | SVD | 22.95 | 706.81 | 732.03 | 856.19 | 1028.59 | 1233.48 | 1110.17 |
| | TADW | 85.27 | 496.39 | 556.72 | 675.35 | 789.50 | 955.53 | 879.97 |
| | ALS | 10.33 | 488.21 | 546.06 | 666.74 | 783.90 | 951.99 | 876.16 |
| K | SVD | 18.55 | 746.77 | 819.15 | 876.92 | 1055.35 | 1260.61 | 1142.24 |
| | TADW | 75.36 | 515.88 | 617.71 | 689.47 | 810.71 | 976.30 | 905.83 |
| | ALS | 19.63 | 507.52 | 607.41 | 678.62 | 804.52 | 972.43 | 901.93 |
| V | SVD | 47.67 | 486.51 | 566.84 | 776.33 | 1031.01 | 1298.15 | 1381.93 |
| | TADW | 21.30 | 351.72 | 433.80 | 598.57 | 784.06 | 994.83 | 1060.45 |
| | ALS | 20.93 | 350.45 | 430.95 | 597.47 | 783.31 | 993.91 | 1060.27 |
| O | SVD | 10.91 | 98.65 | 209.17 | 318.40 | 424.10 | 485.60 | 651.81 |
| | TADW | 7.46 | 63.88 | 139.42 | 214.37 | 284.44 | 329.65 | 387.38 |
| | ALS | 4.06 | 62.47 | 137.24 | 211.74 | 280.30 | 327.61 | 380.27 |
| Gate | SVD | 78.98 | 620.37 | 734.07 | 933.18 | 1267.13 | 1575.09 | 1723.48 |
| | TADW | 53.27 | 480.31 | 579.55 | 740.82 | 985.39 | 1230.11 | 1336.22 |
| | ALS | 50.51 | 478.15 | 576.39 | 737.95 | 981.58 | 1227.42 | 1333.26 |
| Up | SVD | 83.27 | 598.93 | 756.48 | 977.76 | 1274.48 | 1578.90 | 1736.28 |
| | TADW | 54.63 | 451.98 | 588.23 | 766.85 | 979.54 | 1217.99 | 1335.55 |
| | ALS | 52.39 | 451.09 | 586.46 | 765.01 | 977.22 | 1216.31 | 1333.26 |
| Down | SVD | 31.10 | 166.75 | 267.84 | 464.84 | 791.82 | 939.82 | 1715.65 |
| | TADW | 20.79 | 136.29 | 230.51 | 384.25 | 654.92 | 809.53 | 1040.44 |
| | ALS | 19.69 | 131.20 | 222.02 | 372.51 | 632.20 | 782.88 | 968.54 |

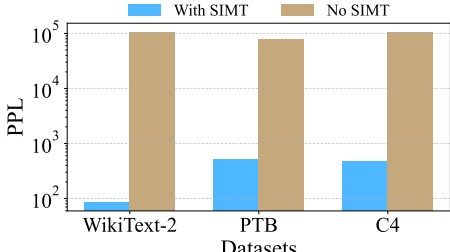

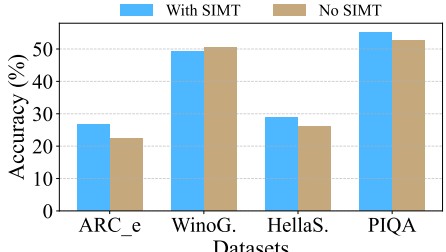

Figure 9: Perplexity comparison of Act-ALS compressed models with SIMT and with "No SIMT".

Figure 10: Accuracy comparison of Act-ALS compressed models with SIMT and with "No SIMT".

## A.7 THE EFFECT OF SIMT

To better understand the role of SIMT, we evaluate the models obtained after the Act-ALS compression step with and without SIMT on both language-modeling and downstream understanding tasks. Figures 9 and 10 report perplexity (PPL) on WikiText-2, PTB and C4, and accuracy on ARC-e, WinoGrande, HellaSwag and PIQA, respectively. The setting denoted as "No SIMT" is implemented by replacing the spectral factor matrices with identity matrices, while the "With SIMT" setting uses spectral factor matrices extracted from the calibration data.

## A.8 ENERGY CONSUMPTION ANALYSIS

To investigate the energy consumption of our pipeline, we break down the runtime and peak GPU memory usage of the SIMT, Act-ALS and UW-MOD stages. For SIMT, we vary the amount of calibration data from 32 to 512 samples and record both wall-clock time and peak GPU memory. As

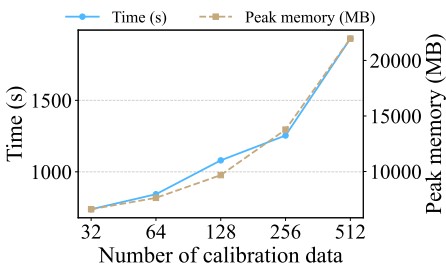

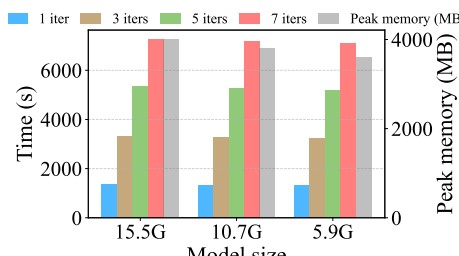

Figure 11: Time and peak memory usage of the SIMT step when varying the number of calibration samples.

Figure 12: Runtime and peak memory usage of the Act-ALS step under different compression scales and numbers of ALS iterations.

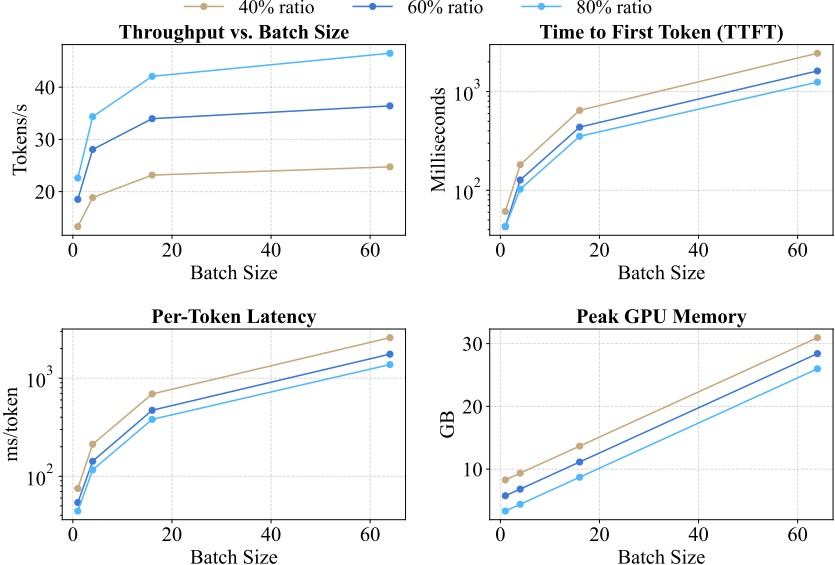

Figure 13: Inference efficiency under 40%, 60%, and 80% compression. Each panel shows one metric versus batch size. Higher compression consistently improves throughput, latency, and memory.

shown in Figure 11, both runtime and peak memory usage increase with the number of calibration samples, which is expected since more calibration data lead to heavier spectral factorization.

For the Act-ALS stage, we further analyze the impact of different compression scales and ALS iterations. Figure 12 plots the model size together with the runtime and peak memory usage under 1, 3, 5 and 7 ALS iterations. We observe that the runtime is mainly dominated by the number of ALS iterations, while the peak memory usage is primarily determined by the compression scale (i.e., the final model size). Moreover, if multiple devices are available, our Act-ALS algorithm naturally supports parallel execution across layers, and in theory its runtime can be reduced by up to a factor equal to the number of layers being compressed.

For the UW-MOD stage, we conduct experiments with 256 calibration samples and vary the number of data loaders. We measure the wall-clock time and peak GPU memory usage for a single epoch. The results, shown in Figure 14, indicate that increasing the number of data loaders consistently reduces the per-epoch runtime, while the peak memory usage gradually increases. Similar to Act-ALS, the UW-MOD stage also naturally supports parallel execution across multiple devices, and in theory its runtime can be accelerated by a factor proportional to the number of devices.

To further quantify the overall adaptation cost, we emphasize that our pipeline always uses the same calibration set throughout, whose size is fixed to 256 samples. In order to compare with the LoRA

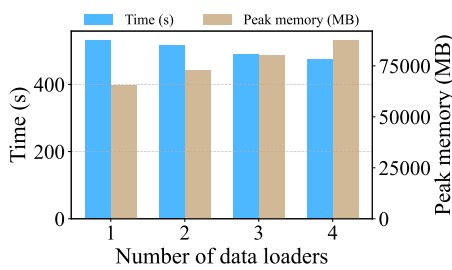
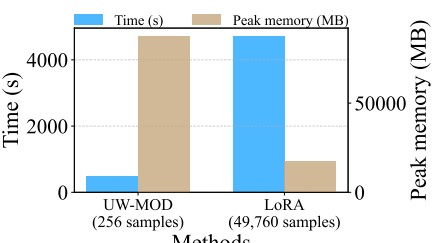

Figure 14: Runtime and peak memory usage of the UW-MOD step under different numbers of data loaders (256 calibration samples).

Figure 15: Per-epoch runtime and peak memory usage of UW-MOD using 256 calibration samples versus LoRA fine-tuning using 49,760 training samples.

Table 10: Performance on generation benchmarks for the original LLaMA-7B model (Original) and its 60% compressed variants. We report scores (in %) on GSM8K (math reasoning), XSum (abstractive summarization), and LAMA (factual knowledge probing). ALS-ActLR denotes our method.

| Method | GSM8K | XSum | LAMA |
|--------|-------|------|------|
| Original | 3.2% | 19.0% | 2.1% |
| Lillama | 0.9% | 3.8% | 0.0% |
| SVD-LLM V2 | 0.6% | 4.1% | 0.8% |
| SoLA | 0.8% | 0.1% | 0.0% |
| LoRKD | 1.2% | 3.5% | 1.3% |
| **ALS-ActLR** | **1.4%** | **4.2%** | **1.8%** |

fine-tuning setup in Section 4.4, we report the per-epoch runtime and peak GPU memory usage for UW-MOD and LoRA, respectively. As shown in Figure 15, UW-MOD only relies on the 256 calibration samples, whereas LoRA is trained on 49,760 fine-tuning examples. It is worth noting that the goal of UW-MOD is to recover the information about the calibration distribution that may be lost in the Act-ALS stage, rather than to perform general-purpose fine-tuning of the model on data beyond the calibration set.

## A.9 SENSITIVITY ANALYSIS

To study the sensitivity of the Act-ALS stage to its regularization hyperparameters, we focus on the ridge term $\tau$ and the proximal term $\rho$ in the ALS objective. We fix the number of ALS iterations to 6 and perform a grid search over $\tau \in \{0.003, 0.03, 0.3, 3\}$ and $\rho \in \{0.003, 0.03, 0.3, 3\}$. For each configuration, we apply Act-ALS compression and report the average perplexity (PPL) and average accuracy across all evaluation benchmarks. As shown in Figures 16 and 17, the ridge term has a stronger impact on performance than the proximal term. When $\tau$ lies in the range $[0.003, 0.03]$, both PPL and accuracy are relatively insensitive, indicating that Act-ALS is fairly robust in this regime. However, once $\tau \geq 0.3$, the compression quality degrades, resulting in higher PPL and lower accuracy compared with smaller values. Our default choice $\tau = 0.003$ and $\rho = 0.003$ yields the best overall performance.

## A.10 THE PERFORMANCE ON GENERATION TASKS

We evaluate three generation benchmarks that cover different abilities: GSM8K (Namburi et al., 2023) for math reasoning, XSum for abstractive summarization, and LAMA (Zhang et al., 2025) for factual knowledge probing. On LLaMA-7B with a 60% compression scale, the results in Table 10 show that ALS-ActLR consistently outperforms other compression baselines on all three benchmarks, indicating that our method also maintains strong performance on generation tasks.

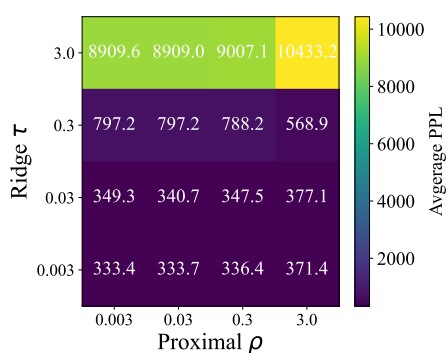
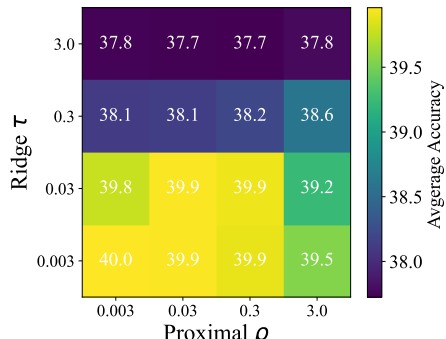

Figure 16: Average PPL of Act-ALS compressed models under different choices of ridge coefficient $\tau$ and proximal coefficient $\rho$ (6 ALS iterations).

Figure 17: Average accuracy of Act-ALS compressed models under different choices of ridge coefficient $\tau$ and proximal coefficient $\rho$ (6 ALS iterations).

## B  LLM USAGE

Large language models were employed solely as assistive tools for language editing and polishing. They were not involved in the design of ALS-ActLR, the theoretical analysis, the implementation of algorithms, or the execution of experiments. All technical contributions, experimental results, and claims were developed and validated independently by the authors, who take full responsibility for the paper's content.

