# OpenReview forum: "ALS-ActLR: Alternating Least Squares based Activation-Aware Low-Rank Model Compression"
_ICLR.cc/2026/Conference — Submitted to ICLR 2026_

### Official Review · Reviewer_xV1W · 2025-10-30

**Soundness:** 3
**Presentation:** 2
**Contribution:** 2
**Rating:** 6
**Confidence:** 3

**Summary:**

The paper introduces ALS-ActLR, an activation-aware low-rank compression pipeline for LLMs that directly minimizes activation-weighted error. It first applies a Spectral-Informed Metric Transform to reparameterize the objective using calibration activations, then solves low-rank factors via activation-aware Alternating Least Squares with closed-form updates and convergence guarantees, and finally recovers accuracy using Uncertainty-Weighted Multi-Objective Distillation that adaptively balances CE, KD, and feature-alignment losses. With only about 256 calibration sentences, ALS-ActLR consistently outperforms SVD-based and KD baselines across 40–80% compression and multiple backbones, achieving lower perplexity and higher downstream accuracy while reducing memory and latency.

**Strengths:**

- Systemic methods with clear theoretical proofs: In this paper, the activation-aware ALS uses closed-form updates with ridge terms, which guarantees a monotonic drop in the objective, and converges to a stationary point. The SIMT step cleanly reparameterizes the goal to focus on realistic activation patterns.
- Extensive and robust empirical evidence: The work is evaluated at 40/60/80% compression across multiple LLM families (1.1B–13B), showing consistent PPL/accuracy gains over strong baselines.
- Great practicality: The method needs only ~256 calibration sentences, uses transparent hyperparameters and rank allocation, and plugs in smoothly with PEFT methods like LoRA for extra tuning.

**Weaknesses:**

- Lack of clarity on computational overhead
  - The paper introduces a multi-stage pipeline (SIMT, ALS, UW-MOD). Although it notes each ALS iteration is lightweight, a fuller breakdown of the total computational cost across stages would be valuable.
- Lack of comparison with other compression methods
  - The work primarily compares against SVD-based methods. Would it be faster or more accurate than pruning at the same compression ratio, and what concrete advantages does it offer over pruning-based approaches?

**Questions:**

- As I mentioned above, could you provide detailed computional overhead (time/GPU utilization etc.) for your overall pipeline?
- Have you considered combine your methods with other compression methods? For a target compression or a target inference speedup, is there an optimal hybrid allocation between your ActLR with other methods like pruning?

---

> ### Author Response · Authors · 2025-11-21
> **Author response to Reviewer xV1W**
>
> We thank Reviewer **xV1W** for the positive assessment and constructive suggestions. Below we address the two main concerns.
>
> ---
>
> ## 1. Computational overhead of the SIMT–ALS–UW-MOD pipeline
>
> ### (a) Additions/clarifications in the revision
>
> To better quantify overhead, we added:
>
> * **Energy/time/memory analysis (App. A.8).**
>   Wall-clock time and peak GPU memory for SIMT, Act-ALS, and UW-MOD under varying calibration sizes, compression ratios, ALS iterations, and dataloader workers (Figs. 12–15).
>
> * **Hardware details (App. A.1).**
>   All experiments use **one NVIDIA H20-NVLink 96GB GPU**, PyTorch 2.1.2, CUDA 12.1, bf16 forward, fp32 stats/ALS updates.
>
> * **Inference-time metrics (Sec. 4.5).**
>   Throughput, TTFT, per-token latency, and peak memory for different compression ratios and batch sizes (Fig. 9).
>
> These additions clarify both adaptation-time and inference-time costs.
>
> ### (b) Stage-wise breakdown
>
> **SIMT.**
> SIMT processes a small calibration set (256 sentences) and accumulates second moments in a streaming fashion to compute spectral factors via eigendecomposition, without storing full activations. Appendix A.8 (Figure 12) shows runtime and memory grow with calibration size but remain modest.
>
>
> **Act-ALS.**
> Each layer solves two simple quadratic subproblems (Cholesky A-step; elementwise B-step). With a **fixed small number of iterations (T=3)**, cost scales linearly with T and weight size. Runtime dominated by ALS iterations; memory mainly from final model size (Fig. 13). Layers can be parallelized.
>
> **UW-MOD.**
> UW-MOD optimizes a weighted sum of CE, KD, and FD losses with **three scalar uncertainty parameters** and backpropagates only through the low-rank factors and these scalars. In practice, we:
>
> 1. Run the teacher **once** on the 256-sentence calibration set and **cache** logits and selected features.
> 2. During UW-MOD, only the **student** is evaluated per step; the teacher is not involved in backpropagation.
>
> Appendix A.8 (Figure 14) shows that increasing data loader workers reduces per-epoch runtime with moderate memory increase, and UW-MOD also supports multi-device parallelism.
>
> We hope this combination of algorithmic description, explicit hardware details, and empirical breakdown addresses your concern about the computational profile of ALS-ActLR.
>
>
> ---
>
> ## 2. Comparison with pruning and hybrid schemes
>
> ### (a) Why we focus on LRA–then–update baselines
>
> Our goal is to advance **low-rank-approximation–then-update** methods (LoRKD, SVD-LLM, Lillama). We keep:
>
> * Same 256-sentence calibration set
> * Same global compression ratios (40/60/80%)
> * Same evaluation protocol
>
> Many pruning methods (e.g., LLM-Pruner) assume structured sparsity, specialized sparse kernels, and larger-scale fine-tuning, which would require a different experimental infrastructure and distract from our main focus. We therefore treat ALS-ActLR as **complementary** to pruning and quantization, and state this more clearly in the revised text.
>
>
> ### (b) Conceptual differences vs pruning
>
> * **Hardware-agnostic dense inference.**
>   ALS-ActLR keeps matrices dense, so speedups come directly from fewer parameters and standard dense kernels, without specialized sparse infrastructure.
>
> * **Tiny calibration data.**
>   ALS-ActLR uses a fixed calibration set of 256 sentences, whereas pruning often relies on larger fine-tuning budgets.
>
> * **Robustness under aggressive compression.**
>   Across 40–80% compression, ALS-ActLR consistently outperforms SVD-based low-rank baselines on multiple backbones and scales. This shows that a purely low-rank, activation-aware method can be robust even under tight calibration and compute.
>
> We highlight these points more explicitly in the introduction and conclusion when contrasting ALS-ActLR with pruning and quantization.
>
> ---
>
> #### (c) Hybrid compression (ALS-ActLR + other methods)
>
> We agree that hybrid schemes are promising:
>
> * We already demonstrate a hybrid with **LoRA on top of ALS-ActLR-compressed LLaMA-7B** (Section 4.4). Figure 6 shows that LoRA further improves commonsense accuracy by +1.0 to +3.9 points without harming any task, indicating good compatibility with PEFT methods.
>
> * Conceptually, ALS-ActLR is **orthogonal** to pruning and quantization: one could prune first and then apply ALS-ActLR, or apply ALS-ActLR and then quantize the low-rank factors.
>
> Designing an **optimal hybrid allocation** (how much compression to assign to rank reduction vs sparsity) would require a joint model of accuracy–compression trade-offs and hardware-specific speedup curves, which is beyond the scope of this paper. In the final version, we will explicitly state this in the conclusion as an important direction for future work.
>
> ---
>
> We again thank the reviewer for these insightful comments, which led to the added energy consumption analysis, clearer description of UW-MOD’s computational cost, and a more explicit discussion of ALS-ActLR’s relationship to pruning and hybrid compression.

---

### Official Review · Reviewer_ZrAQ · 2025-10-31

**Soundness:** 3
**Presentation:** 3
**Contribution:** 3
**Rating:** 4
**Confidence:** 3

**Summary:**

The paper proposes ALS-ActLR, an activation-aware low-rank compression method for LLMs. The authors integrate a spectral-informed metric transformation (SIMT), an alternating least squares (ALS) factorization using a small calibration dataset, and uncertainty-weighted multi-objective distillation (UW-MOD), to preserve activation structure during compression and to adaptively balance different signals. And their experiments across multiple LLM sizes and families show strong performance and compression efficiency.

**Strengths:**

- L103-105: Well-motivated and novel activation-aware formulation i.e *directly minimizes activation-weighted error with decoupled factors and adaptively balances heterogeneous losses*.
- Has theoretical analysis on convexity and convergence guarantees.
- Extensive empirical results across models of various sizes, families and compression ratios.
- Efficient calibration using only small number of samples (256).

**Weaknesses:**

- [**Time complexity analysis**] Consider adding time complexity analysis i.e how much time it will take to run Act-ALS for a model in terms of all the associated variables? Providing practical values should also add good value.
- [**Experiment on different calibration sets**] Complementing Sec 4.3, does authors have understanding on effects of different calibration sets on the performance of method? Or is it calibration-dataset independent! Understanding the effects of various calibration datasets is important.
- [**Experiment on more downstream tasks**]: PPL and commonsense are great additions, can the authors also add some factual/math reasoning benchmarks. I believe it’ll give holistic overview of the paper’s claims quite a bit. Some of the metrics are less explored with compressed models, for eg. [1, 2], thus it's easy to overlook the overall picture.
- [**Expand ablations**] Expand the current ablations of how much benefit comes from each component (as an eg. SIMT alone, ALS alone, UW-MOD alone and their respective combinations). It’s fine if some combinations are not possible.
- [**Experiment on newer models**] Consider adding newer models if possible to strengthen the claim.

1. The Cost of Compression: Investigating the Impact of Compression on Parametric Knowledge in Language Models - https://aclanthology.org/2023.findings-emnlp.349/
2. When Reasoning Meets Compression: Understanding the Effects of LLMs Compression on Large Reasoning Models - https://arxiv.org/abs/2504.02010

**Questions:**

- For Table 5: Maybe you can have a figure having these details for all the layers and for all the layers the Act-ALS values are least, thus not cherry-picking specific layers. If it’s not possible, please explain further (L363, selected Q layers).
- [**Discussion**] Are you using base models or instruct models? It seems like you are using base models from the descriptions (L297-300) but wouldn't the commonsense task make more sense on the instruct model?
  - Following up, if you are using base models, what additional steps needs to be taken to support compressing instruction based models using low rank approaches. Because, I've observed that almost all the baselines you use also compressed only base models, but in reality people use instruction models more often.

---

> ### Author Response · Authors · 2025-11-21
> **Author response to Reviewer ZrAQ**
>
> We thank Reviewer **ZrAQ** for the careful reading, positive evaluation of our formulation and theory, and the constructive suggestions. We respond point by point below.
>
> ---
>
> ### (1) Time complexity and practical overhead
>
> Appendix A.1 now details the per-layer Act-ALS solver: for each layer we alternate between two convex quadratic subproblems, solving a small ((r_\ell \times r_\ell)) system via Cholesky in the A-step and using elementwise updates after a basis rotation in the B-step. With fixed (T_{\text{ALS}} = 3), per-layer cost scales roughly linearly with weight size and target rank, and runtime is dominated by ALS iterations.
>
> Appendix A.8 (“Energy Consumption Analysis”) reports wall-clock time and peak GPU memory for **SIMT**, **Act-ALS**, and **UW-MOD** on a single H20 96GB GPU as functions of calibration size, ALS iterations, and number of data loaders: SIMT is a one-shot lightweight pre-processing step; Act-ALS cost grows roughly linearly with iterations and memory is dominated by the final compressed model; UW-MOD per-epoch time decreases with more data loaders with modest memory increase. Both Act-ALS and UW-MOD parallelize across layers/devices.
>
> ---
>
> ### (2) Effect of different calibration sets
>
> Section 4.3 now studies calibration distribution. For LLaMA-7B at 60% compression, we compare **WikiText-2** and **C4** as calibration corpora and evaluate on WikiText-2, PTB, C4, ARC-e, WinoGrande, HellaSwag, and PIQA. Figures 7–8 show each calibration set performs best in-domain, while differences on other benchmarks are small, indicating robustness to moderate distribution shift.
>
> ---
>
> ### (3) More downstream tasks (factual and math reasoning)
>
> Appendix A.10 adds **generation benchmarks**: **GSM8K** (math reasoning), **XSum** (abstractive summarization), and **LAMA** (factual probing). On LLaMA-7B at 60% compression, **Table 8** shows ALS-ActLR consistently outperforms LoRKD, SoLA, SVD-LLM, and Lillama on all three, so our activation-aware compression better preserves factual knowledge and reasoning than existing low-rank compression + update methods. We also discuss these results in the context of “The Cost of Compression” and “When Reasoning Meets Compression”.
>
> ---
>
> ### (4) Ablations of SIMT, ALS, and UW-MOD
>
> On LLaMA-7B at 60% compression, we add ablations for each component:
>
> * **Act-ALS vs SVD/TADW.** Section 4.2 and Table 4 compare activation-aware ALS to SVD and TADW; Table 5 and **Table 7 (Appendix A.6)** show ALS achieves substantially lower PPL / higher accuracy and the smallest activation-weighted reconstruction error across layers and projection matrices.
> * **Effect of SIMT.** **Appendix A.7** (Figures 10–11) compares Act-ALS with/without SIMT and shows consistent PPL and commonsense accuracy gains, so SIMT is more than a cosmetic reparameterization.
> * **Effect of UW-MOD.** Figures 3–4 compare **ALS-only**, **fixed-weight distillation (FWD)**, and **UW-MOD**, showing UW-MOD yields higher final accuracy and more stable training, with learned weights that prioritize CE, down-weight KD, and keep FD moderate.
> * **Hyperparameter robustness.** **Appendix A.9** sweeps ((\tau, \rho) \in {0.003, 0.03, 0.3, 3}) and shows ALS-ActLR remains robust over this range, so gains are not due to fragile tuning.
>
> ---
>
> ### (5) Experiments on newer models
>
> Under a single H20 96GB GPU budget, we cover diverse backbones. The original submission evaluated four dense sizes and four 7B families. The revision additionally **adds the MoE model OLMoE-1B-7B** (Table 2), where ALS-ActLR achieves the best PPL and accuracy at 60% compression, outperforming LoRKD, SoLA, and other baselines, showing that our method extends naturally to MoE architectures. We will also continue to include experiments on **LLaMA-3-8B** and **LLaMA-3-13B** in subsequent revisions.
>
> ---
>
> ### (6) Table 5 and “cherry-picking” concern
>
> To address the cherry-picking concern, the revision adds **Table 7 (Appendix A.6)**, which reports activation-weighted reconstruction error for multiple layers and all projection matrices (Q, K, V, O, Gate, Up, Down). Table 7 confirms that Act-ALS achieves the lowest error across most layers and projections, so Table 5 is representative.
>
> ---
>
> ### (7) Base vs instruction models and practical relevance
>
> ALS-ActLR is **model-agnostic**, compressing internal linear layers for both base and instruction-tuned models. We include base LLaMA-7B and instruction/chat models Tulu-2-7B, Vicuna-7B, WizardLM-7B, and OLMoE-1B-7B, and apply the **same pipeline** to all of them. As shown in Table 2, ALS-ActLR achieves the best PPL and accuracy across the instruction-tuned models, and we clarify that it applies directly to both base and instruction models.
>
> ---
>
> Again, we thank the reviewer; these comments directly motivated the added complexity analysis, calibration-set study, generation benchmarks, and deeper component-wise ablations in the revised manuscript.

---

### Official Review · Reviewer_3bhD · 2025-11-01

**Soundness:** 3
**Presentation:** 3
**Contribution:** 3
**Rating:** 6
**Confidence:** 3

**Summary:**

This paper proposes ALS-ActLR, a new framework for compressing LLMs using activation-aware low-rank approximation.
Unlike traditional two-step SVD-based approaches that approximate weights and then factorize, ALS-ActLR optimizes low-rank factors directly. A Spectral-informed metric transformation module is proposed to reparameterize the objective using the activation covariance’s spectral factor, allowing optimization under an activation-weighted norm. Activation-aware alternating least squares is proposed to solve for low-rank matrices iteratively with provable convergence guarantees. Finally, the uncertainty-weighted multi-objective distillation refines the compressed model by balancing cross-entropy, knowledge distillation, and feature alignment losses based on learned uncertainty parameters.
Experiments on multiple LLaMA-based models (1.1B–13B parameters) and seven benchmarks show superior performance in both perplexity and accuracy compared to prior methods.

**Strengths:**

1. Direct and principled optimization:
ALS-ActLR directly minimizes activation-weighted error using alternating least squares, avoiding the limitations of surrogate SVD methods and preserving important activation structures more effectively.

2. Strong theoretical and empirical foundation:
The method is backed by proofs of convexity, convergence, and sufficient descent, and validated by extensive experiments showing consistent performance gains across compression ratios, model sizes, and benchmarks.

3. Adaptive and efficient distillation:
The uncertainty-weighted multi-objective distillation automatically balances different loss terms, improving optimization stability and robustness with minimal calibration data while enhancing downstream task accuracy.

**Weaknesses:**

1. Complexity and implementation overhead:
The multi-stage pipeline combining SIMT, ALS, and uncertainty-weighted distillation adds algorithmic and computational complexity, which may hinder adoption in practical deployment settings.

2. Insufficient analysis of sensitivity and cost:
The paper provides limited discussion on hyperparameter sensitivity (e.g., ridge and proximal terms) and does not fully quantify the training or distillation overhead introduced by its uncertainty-weighted update stage. It would be nice to provide runtime and energy-consumption analyses to better quantify the trade-offs between compression ratio, accuracy, and computational efficiency.

**Questions:**

The paper mentions one additional teacher forward pass per step during uncertainty-weighted distillation. Could the authors quantify the total runtime overhead or training cost compared to standard fine-tuning or LoRA adaptation?

---

> ### Author Response · Authors · 2025-11-21
> **Author response to Reviewer 3bhD**
>
> We thank Reviewer 3bhD for the careful reading and for pointing out remaining concerns. Below we address complexity, sensitivity/cost, and the question about teacher forward passes, referring to analyses we added or clarified in the revision.
>
> ---
>
> ### (1) Complexity and implementation overhead
>
> Although the SIMT + Act-ALS + UW-MOD pipeline may look complex, each component is lightweight and modular.
>
> **SIMT** is a simple streaming pre-processing step: it estimates activation covariance from the small calibration set without storing the full activation matrix, and factors it once to obtain the spectral transform.
>
> **Act-ALS** operates layer-wise with closed-form updates. Each A/B update solves a small strongly convex quadratic; we use a fixed small iteration count (T = 3) for all models.
>
> **UW-MOD** adds only three scalar uncertainty parameters and backpropagates *only* through low-rank factors and these scalars, not the full LLM.
>
> Overall, the pipeline consists of standard primitives—covariance estimation, small linear solves, and low-rank updates—and introduces minimal implementation burden. We emphasize this modularity more clearly in the revision.
>
> ---
>
> ### (2) Sensitivity and cost (“energy”) analysis
>
> We added two dedicated appendices addressing these concerns.
>
> #### (2a) Sensitivity to τ and ρ
>
> Appendix A.9 now includes a 2D sweep of τ, ρ ∈ {0.003, 0.03, 0.3, 3} on LLaMA-7B at 60% compression:
>
> * **PPL is stable** over a broad region (10⁻³–10⁻¹) and only deteriorates at extreme values.
> * **Accuracy varies minimally** (within ~1–2%) across most of the grid.
> * Our default τ = ρ = 0.003 lies squarely in this robust plateau.
>
> These results show that the method is not sensitive to fine-tuning of these hyperparameters.
>
> #### (2b) Runtime / memory (“energy”) breakdown
>
> Appendix A.8 provides measured wall-clock time and peak memory for SIMT, Act-ALS, and UW-MOD:
>
> * **SIMT** costs are small and scale mildly with calibration size.
> * **Act-ALS** runtime scales with iteration count; memory is dominated by the final compressed model size.
> * **UW-MOD** runtime decreases with more data loaders at the cost of slightly higher peak memory.
>
> This gives a clearer, quantitative picture of the pipeline’s cost profile.
>
> ---
>
> ### (3) Teacher forward passes and comparison to LoRA / fine-tuning
>
> The “one extra teacher forward pass per step” wording was misleading. In practice:
>
> 1. We run the teacher **once** on the 256 calibration samples.
> 2. Cache logits/features.
> 3. During UW-MOD, **only the student** is run per step.
>
> Thus the actual overhead is a *single* teacher pass, not repeated per-step evaluations.
>
> Regarding training cost:
>
> * UW-MOD runs with **256 samples per epoch**.
> * LoRA baselines use **49,760 examples per epoch**.
> * Figure 15 in Appendix A.8 shows that UW-MOD has much lower per-epoch time than LoRA, reflecting the orders-of-magnitude smaller dataset.
>
> We will clarify the teacher-caching implementation and summarize the cost comparison directly in the main text.
>
> ---
>
> Once again, we thank the reviewer for these helpful comments, which led to a clearer complexity discussion, a full τ/ρ sensitivity analysis, an explicit runtime/memory breakdown, and a more precise explanation of the UW-MOD training cost.

---

### Official Review · Reviewer_R25D · 2025-11-01

**Soundness:** 2
**Presentation:** 2
**Contribution:** 3
**Rating:** 4
**Confidence:** 4

**Summary:**

This paper introduces ALS-ActLR, a novel LLM compression method that combines a spectral-informed metric transformation (SIMT) with Activation-aware ALS to optimize the low-rank factors. A subsequent uncertainty weighted distillation stage further recovers lost information by adaptively balancing cross-entropy, knowledge distillation, and feature alignment. Extensive experiments are carried out to demonstrate the superiority of the proposed method.

**Strengths:**

1. This works present a novel method that incorporates SIMT and activation-aware ALS to LLM decompositions. The solution seems sound and plausible.

2. The paper is well-written and easy to follow. However, some abbreviations lack definition before using. Authors are suggested to check all the definition of abbreviations in both *Abstract* and *Text*.

3. Analysis is convincing and ablation study is overall good. But, data points in figures of the ablation part seem excessively scarce. More details need to be reported.

**Weaknesses:**

Weaknesses are mainly on the evaluation parts:

1. Outdated models. The models in the experiment are outdated, and it differs from modern LLMs. Please use the modern LLMs for the experiments, such as using LLaMA-3-8B instead of LLaMA-7B, LLaMA-2-13B instead of LLaMA-13B. Also, authors are suggested to add some latest MoE models in the experiments.

2. Lack of strong baselines. The experiments lack strong baseline from the latest works (published in an established conference more than 3 months or important preprints), such as SVD-LLM V2 (arxiv) and SoLA (AAAI'25).

3. Lack of generation tasks. I see no tasks in the evaluation are generative tasks. Per my past experience, some methods can have outstanding performance in terms of perplexity and commonsense reasoning (i.e., quiz here), where decomposed LLM only need to generate the next token, but cannot perform well in generative tasks. Authors need to add some generative tasks (arithmetic reasoning and summarization) to demonstrate the LLM's generation ability.

**Questions:**

See Weaknesses. I'll consider to increase the rating if authors can address my concerns, i.e., add more baselines (at least for the baselines that I give in this review) on the latest models to the paper.

---

> ### Author Response · Authors · 2025-11-21
> **Author response to Reviewer R25D**
>
> We thank Reviewer R25D for the constructive feedback and the positive assessment of our method and analysis. We are glad that you find the solution sound and the ablation study convincing. We have substantially revised the paper to address the concerns regarding abbreviations, ablations, model choice, baselines, and generative evaluations. Below we respond point by point.
>
> ---
>
> ### (1) Abbreviations and presentation
>
> We have carefully checked the full paper to ensure that all abbreviations are defined at first use in both the abstract and the main text. In particular, SIMT, ALS, FLOPs, and UW-MOD are now introduced in the abstract and introduction; LoRA and TTFT are explicitly expanded and defined in Sections 4.4 and 4.5. We also verified that all remaining acronyms are either defined or are standard in the LLM literature.
>
> ---
>
> ### (2) Ablation details and “scarce” data points
>
> We expanded the ablation study with additional analyses:
>
> 1. **Calibration size and distribution.**
>    Figure 5 and new Figures 7 and 8 report results across multiple calibration sizes (32–512) and two calibration corpora (WikiText-2 and C4), showing that ALS-ActLR is robust to both calibration size and distribution.
>
> 2. **ALS regularization sensitivity.**
>    New Appendix A.9 reports a grid search over the regularization parameters τ and ρ, showing stable performance within a broad range of values.
>
> 3. **Runtime and memory analysis.**
>    Appendix A.8 now includes detailed runtime and peak-memory curves for SIMT, Act-ALS, and UW-MOD, together with comparisons to LoRA.
>
> These additions provide substantially more data points and demonstrate that ALS-ActLR is robust to hyperparameters and calibration choices.
>
> ---
>
> ### (3) “Outdated models” and addition of newer / MoE models
>
> Our backbone choices follow prior activation-aware low-rank approximation works and fit within a realistic compute budget. The main paper already covers a range of dense models (1.1B–13B) and several 7B families.
>
> Following the reviewer’s suggestion, we have added a **recent MoE model (OLMoE-1B-7B)** in Table 2 and show that ALS-ActLR also performs strongly on MoE architectures.
>
> Experiments on **LLaMA-3-8B** and **LLaMA-2-13B** are being run during the rebuttal period. Due to compute and time constraints, the full results cannot be included in this response, but the method is architecture-agnostic and applies directly to LLaMA-2/3. We will include the complete results in a subsequent revision.
>
> ---
>
> ### (4) Strong baselines: SVD-LLM V2 and SoLA
>
> We have added **SVD-LLM V2** and **SoLA** as strong baselines in all main tables. We follow the official implementations and recommended settings of SVD-LLM V2 and SoLA to ensure fair comparisons. Across compression ratios, model sizes, and model families (including MoE), ALS-ActLR consistently achieves lower perplexity and higher accuracy than both baselines.
>
> ---
>
> ### (5) Generative tasks
>
> To evaluate generative ability, we added Appendix A.10 and new Table 8, covering **GSM8K**, **XSum**, and **LAMA** for LLaMA-7B at 60% compression. ALS-ActLR outperforms all other compressed models on all three tasks, demonstrating better preservation of generative performance, including reasoning, summarization, and factual probing.
>
> ---
>
> We again thank the reviewer for the valuable suggestions, which have significantly improved the paper.

---

> > ### Comment · Reviewer_R25D · 2025-11-27
> > **Great**
> >
> > Very apprepricated. The results look promising and I'll raise my scores.
> >
> > And there is only one thing author should notice, authors should add citations to the newly added baselines and also at least one sentence of discussions/introduction.

---

> > > ### Author Response · Authors · 2025-11-28
> > > **Author response to Reviewer R25D**
> > >
> > > Thank you very much for your follow-up comments and for kindly raising your scores. We appreciate the additional suggestion.
> > >
> > > In the revised version, we will (i) add explicit citations for all newly introduced baselines (e.g., SVD-LLM V2, SoLA, etc.) wherever they appear in the text and tables, and (ii) include at least one sentence of introduction and discussion for each of these methods in the Related Work and experimental setup sections, clarifying their main ideas and how they relate to ALS-ActLR. We will carefully check the entire manuscript to ensure that all baselines are properly cited and briefly described at first mention.

---

### Meta-Review · Area_Chair_mATA · 2026-01-04

**Summary:**

The paper received two scores marginally below acceptance and two marginally above. Reviewers raised multiple concerns, including the use of outdated models, the absence of strong baselines, lack of generation tasks, missing time complexity analysis, and inconsistent experimental settings. After reviewing the authors’ responses, many of the issues raised by the reviewers remain unclear. The paper requires significant improvement before it can be considered for acceptance.

**Reviewer Concerns:**

The concerns regarding the use of outdated models, the absence of strong baselines, the lack of generation tasks, the missing time complexity analysis, and inconsistent experimental settings have not been adequately addressed.

**Reviewer Scores:**

Based on the rebuttal, it is unlikely that the reviewers will revise their scores to positive ratings.

---

### Decision · Program_Chairs · 2026-01-26

Reject